# ANALYZING NEURAL NETWORK BASED GENERATIVE DIFFUSION MODELS VIA CONVEXIFICATION

## ABSTRACT

Diffusion models are becoming widely used in state-of-the-art image, video and audio generation. Score-based diffusion models stand out among these methods, necessitating the estimation of the score function of the input data distribution. In this study, we present a theoretical framework to analyze two-layer neural network-based diffusion models by reframing score matching and denoising score matching as convex optimization. We show that the global optimum of the score matching objective can be attained by solving a simple convex program. Specifically, for univariate training data, we establish that the Langevin diffusion process through the learned neural network model converges in the Kullback-Leibler (KL) divergence to either a Gaussian or a Gaussian-Laplace distribution when the weight decay parameter is set appropriately. Our convex programs alleviate issues in computing the Jacobian and also extends to multidimensional score matching.

## 1 INTRODUCTION

Diffusion model (Sohl-Dickstein et al., 2015) has been invented to tackle the problem of sampling from unknown distribution in machine learning area and is later shown to be able to generate high quality graphs in (Ho et al., 2020). Song et al. (2021) recognize diffusion model as an example of score-based models which exploit Langevine dynamics to produce data from an unknown distribution which only requires estimating the score function of the data distribution. Specifically, the simplest form of Langevine Monte Carlo procedure involves first sampling $x^0$ from an initial distribution, then repeating the following update

$$x^t \leftarrow x^{t-1} + \frac{\epsilon}{2} \nabla_x \log p(x^{t-1}) + \sqrt{\epsilon} z^t$$

where $z^t$ is an independently generated random variable. Here, $\nabla_x \log p(x)$ is known as score function of the distribution $p(x)$ we desire to sample from. It can be shown that under certain conditions (Chewi, 2023), we obtain data distributed according to the target distribution $p(x)$ as $\epsilon$ tends to zero and number of iterations tend to infinity. Langevine dynamics sampling procedure suggests that we can attempt to sample from an unknown distribution as long as we can estimate the score function of this distribution at each data point, which is the key observation in current diffusion models designed for generative tasks. In practice, deep neural networks are trained to minimize the score matching or denoising score matching objective for fitting the score function. However, the theoretical assurances of Langevin sampling do not immediately apply to neural network based score models. Crucially, there is a limited understanding of the distribution of samples produced by these models. Moreover, the standard score matching objective includes the computation of trace of Jacobian of neural network output (Hyvärinen, 2005), introducing computational challenges. In addition, this causes major problems when activation function such as ReLU are used, since the Jacobian of neural network output involves threshold functions which have zero gradient almost everywhere. Thus, gradient-based optimization method faces fundamental problems during training of a neural network to minimize the score matching objective.

Recent development of convex neural network literature studies convex programs that are equivalent to the non-convex neural network training problem in the sense that they have the same optimal value and one can construct a neural network parameter set that achieves global minimum of training problem by solving the corresponding convex program. Motivated by (Ergen et al., 2023), in which the authors show the equivalent convex problems for supervised training objectives involving

deep neural network with threshold activation functions, we derive the convex program for score matching objective which solves the score estimation training problem to global optimality when the neural network model consists of two-layers. We then investigate the convex program and show that when data is univariate and weight decay parameter is set appropriately, the sampled data distribution converges to either Gaussian or Gaussian-Laplace distribution (defined in Theorem 6.2) in KL divergence. We also derive convex programs for the denoising score matching objective. Our theoretical findings are verified by numerical simulations.

## 1.1 CONTRIBUTIONS

- We show that the score matching objective can be transformed to a convex program and solved to global optimality for a two-layer neural network, which bypasses the difficulties faced by gradient-based methods for this specific objective as described in the introduction section.

- When the data is univariate, we fully characterize the distribution of the samples generated by the diffusion model where the score function is learned through a two-layer neural network on arbitrary training data for a certain range of weight decay. We show that the generated samples converge to either Gaussian or Gaussian-Laplace distribution in KL divergence. For multivariate data, we establish connections between neural network based diffusion models and score matching objective used in graphical models.

- We present a convex formulation for the denoising score matching objective, an alternative to traditional score matching, tailored for two-layer neural networks. Additionally, we explore annealed Langevin sampling.

## 2 BACKGROUND

Diffusion model has been shown to be useful in various generative tasks including graph generation (Ho et al., 2020), audio generation (Zhang et al., 2023), and text generation (Wu et al., 2023). Variants of diffusion models such as denoising diffusion implicit model (Song et al., 2022) have been designed to speedup sample generation procedure. The key to score-based diffusion model is the estimation of score function at any data point. In practice, a deep neural network model $s_\theta$ is trained to minimize the score matching objective $\mathbb{E}[\|s_\theta(x) - \nabla_x \log p_{\text{data}}(x)\|_2^2]$ and is used for score function estimation. The score matching objective can be shown to be equivalent up to a constant to

$$\mathbb{E}_{p_{\text{data}}(x)} \left[ \text{tr} \left( \nabla_x s_\theta(x) \right) + \frac{1}{2} \|s_\theta(x)\|_2^2 \right] \tag{1}$$

which is practical since $\nabla_x \log p_{\text{data}}(x)$ is not available. To help alleviate the computation overhead in computing trace of Jacobian in (1) for deep neural network and high dimensional data, sliced score matching (Song et al., 2019) that exploits trace estimation method for trace of Jacobian evaluation and denoising score matching (Vincent, 2011) which considers a perturbed distribution and totally circumvents the computation of trace of Jacobian have been proposed. Score matching method is also studied in graphical model selection (Hyvärinen, 2005; Lin et al., 2016). To demonstrate, with linear neural network, the optimal coefficient that minimizes (1) would give the concentration matrix which is important in modeling correlation among variables.

Note for commonly used activation function such as ReLU, trace of Jacobian involves threshold function which has zero gradient almost everywhere. Therefore, conventional gradient-based optimizers may face difficulties minimizing the training objective. Recent developments in convex neural network literature, originated by (Pilanci & Ergen, 2020) and extended to vector output in (Sahiner et al., 2021), introduced convex programs equivalent to neural network training objectives. Specifically, convex program for training neural network with threshold activation function studied by (Ergen et al., 2023) helps tackle the toughness of applying gradient-based method to training objective which has gradient zero almost everywhere. However, the authors only investigate squared loss objective in their work. In this work we seek to derive a convex program equivalent to score matching objective which solves the training problem globally and always finds an optimal neural network parameter set.

## 3 NOTATION

Here we introduce some notations we will use in later sections. We use $\text{sign}(x)$ to denote the sign function taking value 1 when $x \in [0, \infty)$ and $-1$ otherwise, and $\mathbb{1}$ to denote the 0-1 valued indicator function taking value 1 when the argument is a true Boolean statement. For any vector $x$, $\text{sign}(x)$ and $\mathbb{1}\{x \geq 0\}$ applies elementwise. We denote the pseudoinverse of matrix $A$ as $A^\dagger$. We denote subgradient of a convex function $f : \mathbb{R}^d \to \mathbb{R}$ at $x \in \mathbb{R}^d$ as $\partial f(x) \subseteq \mathbb{R}^d$.

## 4 SCORE MATCHING OBJECTIVE AND NEURAL NETWORK ARCHITECTURES

In this section, we describe the training objective and the neural network architecture we investigate. Let $s_\theta$ denote a neural network parameterized by parameter $\theta$ with output dimension the same as input data dimension which is required for score matching estimation. With $n$ data samples, the empirical version of score matching objective (1) is

$$\mathbf{SM}(s_\theta(x)) = \sum_{i=1}^{n} \text{tr}\left(\nabla_{x_i} s_\theta(x_i)\right) + \frac{1}{2}\|s_\theta(x_i)\|_2^2.$$

The final training loss we consider is the above score matching objective together with weight decay term, which writes

$$\min_\theta \ \mathbf{SM}(s_\theta(x)) + \frac{\beta}{2}\|\theta'\|_2^2, \tag{2}$$

where $\theta' \subseteq \theta$ denotes the parameters to be regularized. Let $m$ denote number of hidden neurons. Consider two-layer neural network architecture of general form as below

$$s_\theta(x) = W^{(2)}\sigma\left(W^{(1)}x + b^{(1)}\right) + Vx + b^{(2)} \tag{3}$$

with activation function $\sigma$, parameter $\theta = \{W^{(1)}, b^{(1)}, W^{(2)}, b^{(2)}, V\}$ and $\theta' = \{W^{(1)}, W^{(2)}\}$ where $x \in \mathbb{R}^d$ is the data matrix, $W^{(1)} \in \mathbb{R}^{m \times d}$ is the first-layer weight, $b^{(1)} \in \mathbb{R}^m$ is the first-layer bias, $W^{(2)} \in \mathbb{R}^{d \times m}$ is the second-layer weight, $b^{(2)} \in \mathbb{R}^d$ is the second-layer bias and $V \in \mathbb{R}^{d \times d}$ is the skip connection coefficient. We will consider network models of the form (3) with ReLU, i.e., $\sigma(t) = (t)_+$, and absolute value, i.e., $\sigma(t) = |t|$ activations and also with or without the skip connection term $Vx$.

### 4.1 UNIVARIATE DATA

We consider training data $x_1, \ldots, x_n \in \mathbb{R}$, and assume that these values are distinct. The following theorem gives the convex program equivalent to the score matching objective (2) for one-dimensional data and for both ReLU and absolute value activation with or without skip connection.

**Theorem 4.1.** *When $\sigma$ is ReLU or absolute value activation and for both the network model with the skip connection ($V \neq 0$) and without ($V = 0$) skip connection, denote the optimal score matching objective value (2) with $s_\theta$ specified in (3) as $p^\star$, when $m \geq m^*$ and $\beta > \beta_0$,*

$$p^* = \min_y \ \frac{1}{2}y^T A^T A y + b^T y + c + d\beta\|y\|_1, \tag{4}$$

*where $m^* = \|y^*\|_0$, and $y^*$ is any optimal solution to (4).*

We specify $\{\beta_0, A, b, c, d\}$ for each of the four following neural network architectures corresponding to $\sigma$ being ReLU or absolute value activation with or without skip connection and analyze the predicted score function accordingly. For sake of page limit, we move the result for neural network with absolute value activation and with skip connection to Appendix A.2.

#### 4.1.1 NEURAL NETWORK TYPE I: RELU ACTIVATION WITH NO SKIP CONNECTION.

Here we consider $\sigma$ to be ReLU activation and $V = 0$, then Theorem 4.1 holds with $\beta_0 = 1, A = [\bar{A}_1, \bar{A}_1, \bar{A}_2, \bar{A}_2] \in \mathbb{R}^{n \times 4n}, b = [1^T C_1, 1^T C_2, -1^T C_3, -1^T C_4]^T \in \mathbb{R}^{4n}, c = 0, d = 1$ where $\bar{A}_1 = \left(I - \frac{1}{n}11^T\right)A_1, \bar{A}_2 = \left(I - \frac{1}{n}11^T\right)A_2$ with $[A_1]_{ij} = (x_i - x_j)_+$ and $[A_2]_{ij} = (-x_i + x_j)_+$,

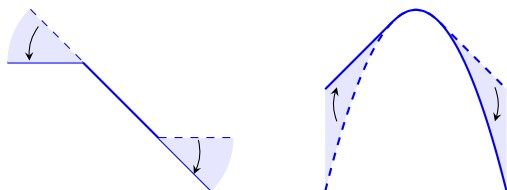

Figure 1: Predicted score function (left) and log of the corresponding probability density (right) for univariate data and type I neural network architecture. The left plot shows all optimal score predictions corresponding to solution to the convex program (4) for weight decay parameter $\beta_1 < \beta \leq n$ and univariate input data of arbitrary distribution.

$[C_1]_{ij} = \mathbb{1}\{x_i - x_j \geq 0\}, [C_2]_{ij} = \mathbb{1}\{x_i - x_j > 0\}, [C_3]_{ij} = \mathbb{1}\{-x_i + x_j \geq 0\}, [C_4]_{ij} = \mathbb{1}\{-x_i + x_j > 0\}$. See Appendix A.3 for proof and reconstruction of optimal parameter set $\theta^\star$. Consider convex program (4), when $\beta > \|b\|_\infty, y = 0$ is optimal and the neural network will always output zero. When $\beta_1 < \beta \leq \|b\|_\infty$ for some threshold $\beta_1$, $y$ is all zero except for the first and the $3n$th entry, which have value $(\beta - n)/2nv + t$ and $(n - \beta)/2nv + t$ for any $|t| \leq \frac{n-\beta}{2nv}$ correspondingly[1]. For any input data point $\hat{x}$, the predicted score $\hat{y}$ is

$$\begin{cases} \hat{y} = \frac{\beta-n}{nv}(\hat{x} - \mu), & x_1 \leq \hat{x} \leq x_n \\ \hat{y} = -(\frac{n-\beta}{2nv} + t)\hat{x} + (\frac{\beta-n}{2nv} + t)x_1 + \frac{n-\beta}{nv}\mu, & \hat{x} < x_1 \\ \hat{y} = (\frac{\beta-n}{2nv} + t)\hat{x} - (\frac{n-\beta}{2nv} + t)x_n + \frac{n-\beta}{nv}\mu. & \hat{x} > x_n \end{cases}$$

where $\mu = \sum_{i=1}^n x_i/n$ denotes the sample mean and $v = \sum_{i=1}^n (x_i - \mu)^2/n$ denotes the sample variance. Figure 1 shows the score function prediction and log of corresponding probability density given by reconstructed optimal neural network via solving convex program (4). Note within sampled data range, the predicted score function aligns with score function of Gaussian distribution parameterized by sample mean $\mu$ and sample variance $v$; outside data range, the predicted score function is a linear interpolation. The integration of score function is always concave in this case, and therefore Langevine dynamics sampling with predicted score function has well-established convergence guarantees (Durmus & Moulines, 2016a; Dalalyan, 2016; Durmus & Moulines, 2016b).

#### 4.1.2 NEURAL NETWORK TYPE II: ABSOLUTE VALUE ACTIVATION WITHOUT SKIP CONNECTION.

When $\sigma$ is absolute value activation and $V = 0$, then Theorem 4.1 holds with $\beta_0 = 1, A = [\bar{A}_1, \bar{A}_1] \in \mathbb{R}^{n \times 2n}, b = [1^T C_1, -1^T C_2]^T \in \mathbb{R}^{2n}, c = 0, d = 1$. where $\bar{A}_1 = (I - \frac{1}{n}11^T) A_1$, $[A_1]_{ij} = |x_i - x_j|, [C_1]_{ij} = \text{sign}(x_i - x_j)$ and $[C_2]_{ij} = \text{sign}(-x_i + x_j)$. See Appendix A.4 for the proof and reconstruction of optimal network parameter set $\theta^\star$. When $\beta > \|b\|_\infty, y = 0$ is optimal and the predicted score is always zero. When $\beta$ is decreased further to some threshold $\beta_2 < n$, i.e., when $\beta_2 < \beta \leq n$, then

$$y = \begin{bmatrix} \frac{\beta-n}{2nv} + t, & 0, & \dots, & 0, & \frac{n-\beta}{2nv} + t \end{bmatrix}^T$$

is optimal with any $t \in \mathbb{R}$ such that $|t| \leq \frac{n-\beta}{2nv}$ where $\mu$ and $v$ denotes the sample mean and sample variance as described in Section 4.1.1 [2]. For any test data $\hat{x}$, the corresponding predicted score $\hat{y}$ is given by

$$\begin{cases} \hat{y} = \frac{\beta-n}{nv}(\hat{x} - \mu), & x_1 \leq \hat{x} \leq x_n \\ \hat{y} = -2t\hat{x} + (\frac{\beta-n}{nv} + 2t)x_1 + \frac{n-\beta}{nv}\mu, & \hat{x} < x_1 \\ \hat{y} = 2t\hat{x} - (\frac{n-\beta}{nv} + 2t)x_n + \frac{n-\beta}{nv}\mu. & \hat{x} > x_n \end{cases} \qquad (5)$$

Surprisingly, the global optimum set is parameterized by the scalar variable $t$, and is not unique. Figure 2 shows the score function prediction and its integration given by reconstructed optimal neural network via solving convex program (4). The score prediction corresponds to score of Gaussian

---

[1]see Appendix A.10 for proof and value of $\beta_1$.

[2]see Appendix A.11 for proof and value of $\beta_2$.

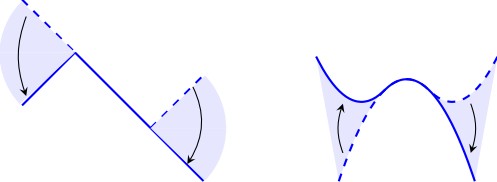

Figure 2: Predicted score function (left) and log of the corresponding probability density (right) for univariate data and type II neural network architecture. The left plot shows all optimal score predictions corresponding to solution to the convex program (4) for weight decay parameter $\beta_2 < \beta \leq n$ and univariate input data of arbitrary distribution.

distribution parameterized by sample mean and sample variance which is the same as the score predicted by type I neural network within the sampled data range as described in Section 4.1.1. The score prediction outside sampled data range is a linear interpolation with a different slope from what is predicted by the type I neural network. This underscores the distinction between absolute value activation and ReLU activation. The corresponding probability density is log-concave only when $t = 0$. Notably, the solution with $t = 0$ corresponds to the unique minimum norm solution of the convex program, highlighting its significance.

### 4.1.3 NEURAL NETWORK TYPE III: ReLU ACTIVATION WITH SKIP CONNECTION

Here we consider $\sigma$ to be ReLU activation with $V \neq 0$, then Theorem 4.1 holds with $\beta_0 = 1, A = B^{\frac{1}{2}}A_1, b = A_1^T(-n\bar{x}/\|\bar{x}\|_2^2) + b_1, c = -n^2/(2\|\bar{x}\|_2^2), d = 2$ where $B = I - P_{\bar{x}}$ with $P_{\bar{x}} = \bar{x}\bar{x}^T/\|\bar{x}\|_2^2$, and $A_1, b_1$ are identical to $A, b$ defined in Section 4.1.2 respectively. Here, $\bar{x}_j := x_j - \sum_i x_i/n$ denotes mean-subtracted data vector. See Appendix A.5 for proof and reconstruction of the optimal parameter set $\theta^\star$. In the convex program (4), $y = 0$ is an optimal solution when $2\beta \geq \|b\|_\infty$. Therefore, following the reconstruction procedure described in Appendix A.5, the corresponding neural network parameter set is given by $\{W^{(1)} = 0, b^{(1)} = 0, W^{(2)} = 0, b^{(2)} = \mu/v, V = -1/v\}$ with $\mu$ and $v$ denotes the sample mean and sample variance as described in Section 4.1.1. For any test data $\hat{x}$, the corresponding predicted score is given by

$$\hat{y} = V\hat{x} + b^{(2)} = -\frac{1}{v}(\hat{x} - \mu),$$

which gives the score function of Gaussian distribution with mean being sample mean and variance being sample variance. Therefore, adding skip connection would change the zero score prediction to a linear function parameterized by sample mean and variance in the large weight decay regime.

### 4.2 EXTENSION TO MULTIVARIATE DATA

The below theorem gives the convex program equivalent to the score matching objective (2) for high-dimensional data and for both ReLU and absolute value activation without skip connection and bias terms. More precisely, $d$ is arbitrary and $b^{(1)} = 0, b^{(2)} = 0, V = 0$. $\sigma$ is either ReLU activation or absolute value activation. For data matrix $X \in \mathbb{R}^{n \times d}$ and any arbitrary vector $u \in \mathbb{R}^d$, We consider the set of diagonal matrices

$$\mathcal{D} := \{\text{diag}(\mathbb{1}\{Xu \geq 0\})\},$$

which takes value $1$ or $0$ along the diagonal that indicates the set of possible arrangement activation patterns for the ReLU activation. Indeed, we can enumerate the set of sign patterns as $\mathcal{D} = \{D_i\}_{i=1}^P$ where $P$ is bounded by

$$P \leq 2r\left(\frac{e(n-1)}{r}\right)^r$$

for $r = \text{rank}(X)$ (Pilanci & Ergen, 2020; Stanley et al., 2004).

**Theorem 4.2.** *When $\sigma$ is ReLU or absolute value activation and $b^{(1)} = 0, b^{(2)} = 0, V = 0, \beta = 0$, denote the optimal score matching objective value (2) with $s_\theta$ specified in (3) as $p^\star$, when $m \geq 2Pd$,*

$$p^\star = \min_{W_j} \frac{1}{2} \left\| \sum_{j=1}^{P} D_j X W_j \right\|_F^2 + \sum_{j=1}^{P} tr(D_j) tr(W_j), \tag{6}$$

where $W_j \in \mathbb{R}^{d \times d}$. $D_j = D_j$ when $\sigma$ is ReLU activation and $D_j = 2D_j - I$ when $\sigma$ is absolute value activation.

*Proof.* see Appendix A.7. □

### 4.2.1 CONNECTION TO GRAPHICAL MODELS

For ReLU activation, denote $\tilde{X} = [D_1 X, \ldots, D_P X], \tilde{V} = [tr(D_1)I, tr(D_2)I, \ldots, tr(D_P)I], W = [W_1, \ldots, W_P]^T$, then the convex program (6) can be written as

$$\min_W \frac{1}{2} \|\tilde{X}W\|_F^2 + \langle \tilde{V}, W \rangle. \tag{7}$$

When the optimal value is finite, e.g., $\tilde{V} \in \text{range}(\tilde{X}^T \tilde{X})$, an optimal solution to (7) is given by

$$W = (\tilde{X}^T \tilde{X})^\dagger \tilde{V}$$

$$= \begin{bmatrix} \sum_{k \in S_{11}} X_k X_k^T & \sum_{k \in S_{12}} X_k X_k^T & \cdots \\ \sum_{k \in S_{21}} X_k X_k^T & \sum_{k \in S_{22}} X_k X_k^T & \cdots \\ & \cdots & \end{bmatrix}^\dagger \begin{bmatrix} tr(D_1)I \\ tr(D_2)I \\ \vdots \\ tr(D_P)I \end{bmatrix},$$

where $S_{ij} = \{k : X_k^T u_i \geq 0, X_k^T u_j \geq 0\}$ and $u_i$ is the generator of $D_i = \text{diag}(\mathbb{1}\{X u_i \geq 0\})$.

*Remark* 4.3. Note that the above model can be seen as a piecewise empirical covariance estimator which partitions the space with hyperplane arrangements. When $P = 1$, $D_1 = I$ and $X^T X$ is invertible, then $(\tilde{X}^T \tilde{X})^\dagger$ reduces to the empirical precision matrix which models the correlation between different data points. This was observed in the application of score matching objective in graphical models (Hyvärinen, 2005; Lin et al., 2016). Here, we obtain a more expressive model with the non-linear neural network through data partitioning.

## 5 DENOISING SCORE MATCHING

To tackle the difficulty in computation of trace of Jacobian required in score matching objective (1), denoising score matching ((Vincent, 2011)) first perturbs data point with a predefined noise distribution and then estimates the score of the perturbed data distribution. When the noise distribution is chosen to be standard Gaussian, for some $\epsilon > 0$, the objective is equivalent to

$$\min_\theta \mathbb{E}_{x \sim p(x)} \mathbb{E}_{\delta \sim \mathcal{N}(0,I)} \left\| s_\theta(x + \epsilon\delta) - \frac{\delta}{\epsilon} \right\|_2^2,$$

and the empirical version is given by

$$\mathbf{DSM}(s_\theta) = \sum_{i=1}^{n} \frac{1}{2} \left\| s_\theta(x_i + \epsilon\delta_i) - \frac{\delta_i}{\epsilon} \right\|_2^2, \tag{8}$$

where $\{x_i\}_{i=1}^n$ are samples from $p(x)$ and $\{\delta_i\}_{i=1}^n$ are samples from standard Gaussian. The final training loss we consider is the above score matching objective together with weight decay term, which writes

$$\min_\theta \mathbf{DSM}(s_\theta(x)) + \frac{\beta}{2} \|\theta'\|_2^2, \tag{9}$$

where $\theta' \subseteq \theta$ denotes the parameters to be regularized. Note (9) circumvents the computation of trace of Jacobian and is thus more applicable for training tasks in large data regime. One drawback is that optimal $s_\theta$ that minimizes (9) measures score function of the perturbed data and is only close to original data distribution when noise is small enough. We consider the same neural network architecture and types described in Section 4 below.

## 5.1 UNIVARIATE DATA

The following theorem gives the convex program equivalent to the score matching objective (9) for one-dimensional data and for both ReLU and absolute value activation with or without skip connection. Let $l$ denotes the label vector, i.e, $l = [\delta_1/\epsilon, \delta_2/\epsilon, \ldots, \delta_n/\epsilon]^T$.

**Theorem 5.1.** *When $\sigma$ is ReLU or absolute value activation and for both the network model with the skip connection ($V \neq 0$) and without ($V = 0$) skip connection, denote the optimal score matching objective value (9) with $s_\theta$ specified in (3) as $p^\star$, when $\beta > 0$ and $m \geq m^*$,*

$$p^\star = \min_y \quad \frac{1}{2}\|Ay + b\|_2^2 + d\beta\|y\|_1, \tag{10}$$

where $m^* = \|y^*\|_0$, and $y^*$ is any optimal solution to (10).

We specify $\{A, b, d\}$ for each of the four neural network architecture corresponding to $\sigma$ being ReLU or absolute value activation with or without skip connection. When $\sigma$ is ReLU activation and $V = 0$, Theorem 5.1 holds with $A = [\bar{A}_1, \bar{A}_2] \in \mathbb{R}^{n \times 2n}, b = \bar{l} \in \mathbb{R}^n, d = 1$ where $\bar{A}_1 = \left(I - \frac{1}{n}11^T\right) A_1, \bar{A}_2 = \left(I - \frac{1}{n}11^T\right) A_2$ with $[A_1]_{ij} = (x_i - x_j)_+$ and $[A_2]_{ij} = (-x_i + x_j)_+$. $\bar{l}_j = l_j - \sum_i l_i/n$ is the mean-subtracted label vector. When $\sigma$ is absolute value activation and $V = 0$, the above theorem holds with $A = \left(I - \frac{1}{n}11^T\right) A_3 \in \mathbb{R}^{n \times n}, b = \bar{l} \in \mathbb{R}^n, d = 1$ where $[A_3]_{ij} = |x_i - x_j|$. When $\sigma$ is the ReLU activation and $V \neq 0$, the above theorem holds with $A = B^{\frac{1}{2}} \left(I - \frac{1}{n}11^T\right) A_3 \in \mathbb{R}^{n \times n}, b = B^{\frac{1}{2}}\bar{l} \in \mathbb{R}^n, d = 2$ where $B = I - P_{\bar{x}}$ with $P_{\bar{x}} = \bar{x}\bar{x}^T/\|\bar{x}\|_2^2$ and $\bar{x}$ being the mean-subtracted data vector as defined in Section 4.1.3. When $\sigma$ is absolute value activation and $V \neq 0$, the above theorem holds with the same $A, b$ as for ReLU activation with skip connection and $d = 1$. See Appendix A.8 for proof and reconstruction of optimal neural network parameter set $\theta^\star$.

## 5.2 EXTENSION TO HIGH-DIMENSIONAL DATA

The below theorem gives the convex program equivalent to the score matching objective (9) for high-dimensional data and for both ReLU and absolute value activation without skip connection and bias terms. Namely, $d$ is arbitrary and $b^{(1)} = 0, b^{(2)} = 0, V = 0$. $\sigma$ is either ReLU or absolute value activation. Let $L \in \mathbb{R}^{n \times d}$ denote the label matrix, i.e., $L_i = \delta_i/\epsilon$, and $\mathcal{D} = \{D_i\}_{i=1}^P$ be the arrangement activation patterns for ReLU activation as defined in Section 4.2.

**Theorem 5.2.** *When $\sigma$ is ReLU or absolute value activation and $b^{(1)} = 0, b^{(2)} = 0, V = 0, \beta = 0$, denote the optimal score matching objective value (9) with $s_\theta$ specified in (3) as $p^\star$, when $m \geq 2Pd$,*

$$p^\star = \min_{W_j} \frac{1}{2} \left\| \sum_{j=1}^P D_j X W_j - L \right\|_F^2 \tag{11}$$

where $W_j \in \mathbb{R}^{d \times d}$. $D_j = D_j$ when $\sigma$ is ReLU activation and $D_j = 2D_j - I$ when $\sigma$ is absolute value activation.

*Proof.* see Appendix A.9. □

## 6 ALGORITHM AND CONVERGENCE RESULT

Strong convergence guarantees for the Langevin Monte Carlo method are often contingent upon the log-concavity of the target distribution. Notably, in Section 4.1 we analyze the predicted score function for different neural network architecture in certain ranges of weight decay. Some of these score functions correspond to log concave distributions and thus we can exploit existing convergence results for Langevine dynamics to derive the convergence of the diffusion model with a neural network based score function. Here, we follow notations used in Section 4.1. Algorithms of score matching and Langevine sampling are given in Algorithm 1 and 2 respectively. To the best of our knowledge, prior to our study, there had been no characterization of the sample distribution generated by Algorithm 2 when the score model is trained using Algorithm 1.

| **Algorithm 1** Score Matching | **Algorithm 2** Langevine Monte Carlo |
|---|---|

**Input:** training data $x_1, \ldots, x_n \in \mathbb{R}^d$
minimize

$$\sum_{i=1}^{n} \frac{1}{2} s_\theta^2(x_i) + \nabla_\theta s_\theta(x_i) + \frac{\beta}{2} \|\theta'\|_2^2$$

**Initialize:** $x^0 \sim \mu_0(x)$
**for** $t = 1, 2, ..., T$ **do**
  $z^t \sim \mathcal{N}(0, 1)$
  $x^t \leftarrow x^{t-1} + \frac{\epsilon}{2} s_\theta(x^{t-1}) + \sqrt{\epsilon} z^t$
**end for**

**Theorem 6.1.** *When $s_\theta$ is of neural network type III and IV and $\beta > \|b\|_\infty$, let $\pi$ denote Gaussian distribution with mean $\mu$ and variance $v$. For any $\tau \in [0, 1]$, if we take step size $\epsilon \asymp 2\tau^2 v$, then for the mixture distribution $\overline{\mu} = T^{-1} \sum_{t=1}^{T} x^t$ and $\overline{\mu}' = T^{-1} \sum_{t=T+1}^{2T} x^t$, it holds that $W_2(\overline{\mu}, \pi) \leq \tau$ and $\sqrt{KL(\overline{\mu}' \| \pi)} \leq \tau$ after*

$$O\left(\frac{1}{\tau^2} \log \frac{W_2(\mu_0, \pi)}{\tau}\right) \quad \textit{iterations}$$

*Proof.* see Appendix A.12.1. □

**Theorem 6.2.** *When $s_\theta$ is of neural network type II and corresponds to the min-norm solution to the corresponding convex program (4) and $\beta_2 < \beta \leq n$, let $\pi^1$ denote the Gaussian-Laplace distribution (defined below). Let $L = (n - \beta) \max(|x_1 - \mu|, |x_n - \mu|)/nv$. For any $\tau > 0$, if we take step size $\epsilon \asymp 2\tau^2/L^2$, then for the mixture distribution $\overline{\mu} = T^{-1} \sum_{t=1}^{T} x^t$, it holds that $\sqrt{KL(\overline{\mu} \| \pi^1)} \leq \tau$ after*

$$O\left(\frac{L^2 W_2^2(\mu_0^+, \pi^1)}{\tau^4}\right) \quad \textit{iterations},$$

where $\mu_0^+ = x^0 + \epsilon s_\theta(x^0)/2$ and $\pi^1$ satisfies

$$\pi^1 \propto \begin{cases} \exp(\frac{\beta-n}{nv}(x_1 - \mu)x + \frac{n-\beta}{2nv}x_1^2), & x < x_1, \\ \exp(\frac{\beta-n}{2nv}x^2 - \frac{\mu(\beta-n)}{nv}x), & x_1 \leq x \leq x_n, \\ \exp(\frac{\beta-n}{nv}(x_n - \mu)x + \frac{n-\beta}{2nv}x_n^2), & x \geq x_n. \end{cases}$$

*Proof.* see Appendix A.12.2. □

## 7 NUMERICAL RESULTS

We now present our simulation results. See Appendix A.14 for additional results. For univariate data and two-layer neural network with ReLU activation and without skip connection, Figure 3 shows our simulation results for score matching tasks. We take $n = 500$ data points sampled from standard Gaussian and we take weight decay parameter $\beta = \|b\|_\infty - 1$.[3] The left plot in Figure 3 shows the training loss where the dashed blue line is the objective value obtained by optimal neural network reconstructed from our derived convex program (4). For non-convex neural network training, we run 10 trials with random parameter initiation and use Adam as optimizer with step size $1e - 2$. We train for 500 epochs. The result shows that our convex program solves the training problem globally and stably. The gap between the non-convex training loss and the objective value obtained by our reconstructed optimal neural network can be caused by the non-smoothness in the training objective. The middle plot in Figure 3 is the score prediction given by optimal reconstructed neural network, which confirms our derived score function (5). The right plot shows the histogram for running Langevine dynamics sampling in Algorithm 2 with $10^5$ data points and $T = 500$ iterations, we take $\mu_0$ to be uniform distribution from $-10$ to $10$ and $\epsilon = 1$.

Figure 4 shows our simulation results for denoising score matching tasks. The left plot in Figure 4 shows denoising score matching training loss where the dashed blue line indicates the loss of optimal neural network reconstructed from our derived convex program (10). We take weight decay parameter $\beta = 0.5$, $n = 1000$ data points with standard Gaussian distribution, and noise level

---

[3]see Section 4 for definition of $b$.

$\epsilon = 0.1$ in (8). The non-convex training uses Adam with step size $1e-2$ and takes 200 epochs. We run 10 trials. The notable gap between non-convex training loss and the reconstructed optimal neural network loss reveals that our convex program solves the training problem globally and stably. The middle plot shows the histogram for samples generated via annealed Langevine process (see Appendix A.13 for algorithm, with $L = 10, T = 10, \epsilon_0 = 2e-5, [\sigma_1, \ldots, \sigma_L]$ being the uniform grid from 1 to 0.01, and $\mu_0$ being uniform distribution from $-1$ to 1) where the non-convex trained neural network is used as the score estimator and the right plot shows the same histogram with reconstructed optimal neural network as score estimator. The right histogram is closer to histogram of standard Gaussian samples compared to the middle histogram, which reveals the superiority of our reconstructed optimal neural network over non-convex trained neural network in score prediction.

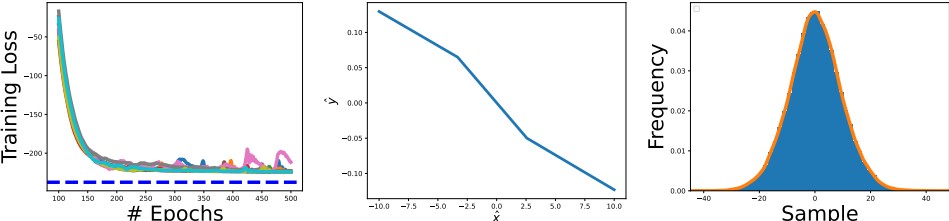

Figure 3: Simulation results for score matching tasks with type I neural network. The left plot shows training loss where the dashed blue line indicates loss of neural network reconstructed from convex program (4). The middle plot shows score prediction from reconstructed optimal neural network. The right plot shows sampling histogram via Langevine process with reconstructed optimal neural network as score estimator. The ground truth distribution is standard Gaussian.

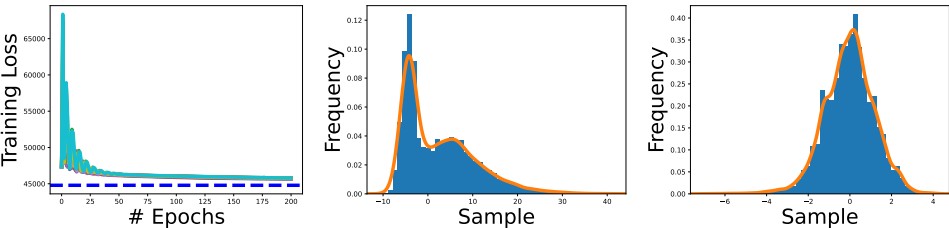

Figure 4: Simulation results for denoising score matching tasks with type I neural network. The left plot shows training loss where the dashed blue line indicates loss of neural network reconstructed from convex program (10). The middle plot shows sampling histogram via annealed Langevine process with non-convex trained neural network as score predictor. The right plot shows sampling histogram via annealed Langevine process with reconstructed optimal neural network as score predictor. The ground truth distribution is standard Gaussian, which is recovered by our model.

# 8 CONCLUSION

In this work, we analyze neural network based diffusion models from the lens of convex optimization. We derive an equivalent convex program for two-layer neural networks trained using the score matching objective, which solves the problem globally and bypasses the difficulty of using gradient-based optimizers due to the Jacobian terms. We also derive the convex program for denoising score matching objective. When data is univariate, we find the optimal set of the convex program for the score matching objective for certain weight decay range, and show that for arbitrary data distributions, the neural-network-learned score function is piecewise linear and can always be parameterized by sample mean and sample variance. We established convergence results for Langevine sampling with neural-network-learned score function.

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

# A APPENDIX

## A.1 LEMMAS

**Lemma A.1.** *The below constraint set is strictly feasible only when $\beta > 1$.*

$$\begin{cases} |z^T(x - 1x_i)_+ - 1^T \mathbb{1}\{x - 1x_i \geq 0\}| \leq \beta \\ |z^T(x - 1x_i)_+ - 1^T \mathbb{1}\{x - 1x_i > 0\}| \leq \beta \\ |z^T(-x + 1x_i)_+ + 1^T \mathbb{1}\{-x + 1x_i \geq 0\}| \leq \beta \qquad \forall i = 1, \cdots, n \\ |z^T(-x + 1x_i)_+ + 1^T \mathbb{1}\{-x + 1x_i > 0\}| \leq \beta \\ z^T 1 = 0 \end{cases}$$

*Proof.* Consider without loss of generality that $x_1 < x_2 < \cdots < x_n$. Let $k = -\sum_{j=1}^{m} z_j(x_j - x_i) + m$ for some $1 \leq m \leq n$, the first four constraints with $i = m$ are then $|z^T x - (n + 1) + k|, |z^T x - (n + 1) + k + 1|, |k|, |k - 1|$. When $i = n$, the first constraint is $\beta \geq 1$. Thus $\beta > 1$ is necessary for the constraint set to be strictly feasible. Since we can always find $z^\star$ satisfying

$$\begin{cases} x^T z^\star = n \\ 1^T z^\star = 0 \\ (x - 1x_i)_+^T z^\star - 1^T \mathbb{1}\{x - 1x_i \geq 0\} = 0 \quad \forall i = 2, \cdots, n - 1 \end{cases}$$

Note such $z^\star$ satisfies all constraints in the original constraint set when $\beta > 1$. Therefore when $\beta > 1$, the original constraint is strictly feasible. $\qquad\square$

**Lemma A.2.** *The below constraint set is strictly feasible only when $\beta > 1$.*

$$\begin{cases} |z^T|x - 1x_i| - 1^T \text{sign}(x - 1x_i)| \leq \beta \\ |z^T| - x + 1x_i| + 1^T \text{sign}(-x + 1x_i)| \leq \beta \quad \forall i = 1, \cdots, n \\ z^T 1 = 0 \end{cases}$$

*Proof.* Consider without loss of generality that $x_1 < x_2 < \cdots < x_n$. Then taking $i = 1$ and $n$ in the first constraint gives $|z^T x - n| \leq \beta$ and $|z^T x - n + 2| \leq \beta$. It's necessary to have $\beta > 1$ and $z^T x = n - 1$ to have both constraints strictly satisfiable. Since we can always find $z^\star$ satisfying the below linear system

$$\begin{cases} x^T z^\star = n - 1 \\ 1^T z^\star = 0 \\ |x - 1x_i|^T z^\star - 1^T \text{sign}(x - 1x_i) = -1 \quad \forall i = 2, \cdots, n - 1 \end{cases}$$

Note such $z^\star$ also satisfies

$$|| - x + 1x_i|^T z^\star + 1^T \text{sign}(-x + 1x_i)| \leq 1$$

Therefore when $\beta > 1$, the original constraint set is strictly feasible. $\qquad\square$

**Lemma A.3.** *The below constraint set is strictly feasible only when $\beta > 2$.*

$$\begin{cases} |z^T|x - 1x_i| - 1^T \text{sign}(x - 1x_i)| \leq \beta \\ |z^T| - x + 1x_i| + 1^T \text{sign}(-x + 1x_i)| \leq \beta \\ z^T 1 = 0 \qquad\qquad\qquad\qquad\qquad\qquad \forall i = 1, \cdots, n \\ z^T x = n \end{cases}$$

*Proof.* Consider without loss of generality that $x_1 < x_2 < \cdots < x_n$. Then taking $i = n$ in the first constraint gives $| - n + (n - 2)| \leq \beta$, which indicates that $\beta > 2$ is necessary for the constraint set to be strictly feasible. Since we can always find $z^\star$ satisfying

$$\begin{cases} x^T z^\star = n \\ 1^T z^\star = 0 \\ |x - 1x_i|^T z^\star - 1^T \text{sign}(x - 1x_i) = 0 \quad \forall i = 2, \cdots, n - 1 \end{cases}$$

Note such $z^\star$ also satisfies

$$|| - x + 1x_i|^T z^\star + 1^T \text{sign}(-x + 1x_i)| \leq 2$$

Therefore when $\beta > 2$, the original constraint set is strictly feasible. $\qquad\square$

## A.2 Neural Network type IV: Absolute value activation with skip connection.

We consider $\sigma$ to be absolute value activation with $V \neq 0$, then Theorem 4.1 holds with the same $A, b, c$ as in Section 4.1.3 and with $\beta_0 = 2, d = 1$. See Appendix A.6 for proof and reconstruction of optimal parameter set $\theta^\star$. Consider convex program (4), when $\beta > \|b\|_\infty, y = 0$ is optimal. Following the reconstruction procedure described in Appendix A.6, the corresponding neural network parameter set is given by $\{W^{(1)} = 0, b^{(1)} = 0, W^{(2)} = 0, b^{(2)} = \mu/v, V = -1/v\}$ with $\mu$ and $v$ denotes the sample mean and sample variance as described in Section 4.1.1. For any testing data $\hat{x}$, the corresponding predicted score is given by

$$\hat{y} = V\hat{x} + b^{(2)} = -\frac{1}{v}(\hat{x} - \mu),$$

which is the score function of Gaussian distribution with mean being sample mean and variance being sample variance, just as the case for $\sigma$ being ReLU activation and $V \neq 0$ described in Section 4.1.3.

## A.3 Proof for Theorem 4.1: Neural Network Type I

*Proof.* Consider data $x \in \mathbb{R}^n$, then the score matching objective is reduced to

$$p^\star = \min_{w,\alpha,b} \frac{1}{2}\left\|\sum_{j=1}^m (xw_j + 1b_j)_+ \alpha_j + 1b_0\right\|_2^2 + 1^T\left(\sum_{j=1}^m w_j\alpha_j \mathbb{1}\{xw_j + 1b_j \geq 0\}\right) + \frac{1}{2}\beta\sum_{j=1}^m (w_j^2 + \alpha_j^2).$$

According to Lemma 2 in Pilanci & Ergen (2020), after rescaling, the above problem is equivalent to

$$\min_{\substack{w,\alpha,b \\ |w_j|=1}} \frac{1}{2}\left\|\sum_{j=1}^m (xw_j + 1b_j)_+ \alpha_j + 1b_0\right\|_2^2 + 1^T\left(\sum_{j=1}^m w_j\alpha_j \mathbb{1}\{xw_j + 1b_j \geq 0\}\right) + \beta\sum_{j=1}^m |\alpha_j|,$$

which can be written as

$$\min_{\substack{w,\alpha,b,r_1,r_2 \\ |w_j|=1}} \quad \frac{1}{2}\|r_1\|_2^2 + 1^T r_2 + \beta\sum_{j=1}^m |\alpha_j|$$

$$\text{s.t.} \quad r_1 = \sum_{j=1}^m (xw_j + 1b_j)_+ \alpha_j + 1b_0$$

$$r_2 = \sum_{j=1}^m w_j\alpha_j \mathbb{1}\{xw_j + 1b_j \geq 0\}.$$

The dual problem writes

$$d^\star = \max_{z_1,z_2} \min_{\substack{w,\alpha,b,r_1,r_2 \\ |w_j|=1}} \frac{1}{2}\|r_1\|_2^2 + 1^T r_2 + \beta\sum_{j=1}^m |\alpha_j| + z_1^T\left(r_1 - \sum_{j=1}^m (xw_j + 1b_j)_+ \alpha_j - 1b_0\right)$$

$$+ z_2^T\left(r_2 - \sum_{j=1}^m w_j\alpha_j \mathbb{1}\{xw_j + 1b_j \geq 0\}\right),$$

which gives a lower bound of $p^\star$. Minimizing over $r_1, r_2, \alpha_j$ gives

$$\max_z \min_{b_0} -\frac{1}{2}\|z\|_2^2 - b_0 z^T 1$$

$$\text{s.t. } |z^T(xw_j + 1b_j)_+ - w_j 1^T \mathbb{1}\{xw_j + 1b_j \geq 0\}| \leq \beta, \quad \forall |W_j| = 1, \forall b_j.$$

For the constraints to hold, we must have $z^T 1 = 0$ and $b_j$ takes values over $x_j$'s. The above is equivalent to

$$\max_{z} \quad -\frac{1}{2}\|z\|_2^2$$

$$\text{s.t.} \begin{cases} |z^T(x - 1x_i)_+ - 1^T \mathbb{1}\{x - 1x_i \geq 0\}| \leq \beta \\ |z^T(x - 1x_i)_+ - 1^T \mathbb{1}\{x - 1x_i > 0\}| \leq \beta \\ |z^T(-x + 1x_i)_+ + 1^T \mathbb{1}\{-x + 1x_i \geq 0\}| \leq \beta \qquad \forall i = 1, \ldots, n, \\ |z^T(-x + 1x_i)_+ + 1^T \mathbb{1}\{-x + 1x_i > 0\}| \leq \beta \\ z^T 1 = 0 \end{cases} \tag{12}$$

According to Lemma A.1, when $\beta > 1$, the constraints in (12) are strictly feasible, thus Slater's condition holds and the dual problem writes

$$d^\star = \min_{\substack{z_0, \ldots, z_7, z_8 \\ \text{s.t.} z_0, \ldots, z_7 \geq 0}} \max_{z} -\frac{1}{2}\|z\|_2^2 + \sum_{i=1}^{n} z_{0i} \left( z^T(x - 1x_i)_+ - 1^T \mathbb{1}\{x - 1x_i \geq 0\} + \beta \right)$$

$$+ \sum_{i=1}^{n} z_{1i} \left( -z^T(x - 1x_i)_+ + 1^T \mathbb{1}\{x - 1x_i \geq 0\} + \beta \right) + \sum_{i=1}^{n} z_{2i} \left( z^T(x - 1x_i)_+ - 1^T \mathbb{1}\{x - 1x_i > 0\} + \beta \right)$$

$$+ \sum_{i=1}^{n} z_{3i} \left( -z^T(x - 1x_i)_+ + 1^T \mathbb{1}\{x - 1x_i > 0\} + \beta \right) + \sum_{i=1}^{n} z_{4i} \left( z^T(-x + 1x_i)_+ + 1^T \mathbb{1}\{-x + 1x_i \geq 0\} + \beta \right)$$

$$+ \sum_{i=1}^{n} z_{5i} \left( -z^T(-x + 1x_i)_+ - 1^T \mathbb{1}\{-x + 1x_i \geq 0\} + \beta \right) + \sum_{i=1}^{n} z_{6i} \left( z^T(-x + 1x_i)_+ + 1^T \mathbb{1}\{-x + 1x_i > 0\} + \beta \right)$$

$$+ \sum_{i=1}^{n} z_{7i} \left( -z^T(-x + 1x_i)_+ - 1^T \mathbb{1}\{-x + 1x_i > 0\} + \beta \right) + z_8 z^T 1,$$

which is equivalent to

$$\min_{\substack{z_0, \ldots, z_7, z_8 \\ \text{s.t.} z_0, \ldots, z_7 \geq 0}} \max_{z} -\frac{1}{2}\|z\|_2^2 + e^T z + f,$$

where $e = \sum_{i=1}^{n} z_{0i}(x - 1x_i)_+ - \sum_{i=1}^{n} z_{1i}(x - 1x_i)_+ + \sum_{i=1}^{n} z_{2i}(x - 1x_i)_+ - \sum_{i=1}^{n} z_{3i}(x - 1x_i)_+ + \sum_{i=1}^{n} z_{4i}(-x + 1x_i)_+ - \sum_{i=1}^{n} z_{5i}(-x + 1x_i)_+ + \sum_{i=1}^{n} z_{6i}(-x + 1x_i)_+ - \sum_{i=1}^{n} z_{7i}(-x + 1x_i)_+ + 1z_8$ and $f = -\sum_{i=1}^{n} z_{0i} 1^T \mathbb{1}\{x - 1x_i \geq 0\} + \sum_{i=1}^{n} z_{1i} 1^T \mathbb{1}\{x - 1x_i \geq 0\} - \sum_{i=1}^{n} z_{2i} 1^T \mathbb{1}\{x - 1x_i > 0\} + \sum_{i=1}^{n} z_{3i} 1^T \mathbb{1}\{x - 1x_i > 0\} + \sum_{i=1}^{n} z_{4i} 1^T \mathbb{1}\{-x + 1x_i \geq 0\} - \sum_{i=1}^{n} z_{5i} 1^T \mathbb{1}\{-x + 1x_i \geq 0\} + \sum_{i=1}^{n} z_{6i} 1^T \mathbb{1}\{-x + 1x_i > 0\} - \sum_{i=1}^{n} z_{7i} 1^T \mathbb{1}\{-x + 1x_i > 0\} + \beta(\sum_{i=0}^{7} \|z_i\|_1)$. Maximizing over $z$ gives

$$\min_{\substack{z_0, \ldots, z_7, z_8 \\ \text{s.t.} z_0, \ldots, z_7 \geq 0}} \frac{1}{2}\|e\|_2^2 + f,$$

Simplifying to get

$$\min_{y_0, y_1, y_2, y_3, y_4} \frac{1}{2} \left\| A_1(y_0 + y_1) + A_2(y_2 + y_3) + 1y_4 \right\|_2^2 + 1^T C_1 y_0 - 1^T C_3 y_2$$

$$+ 1^T C_2 y_1 - 1^T C_4 y_3 + \beta(\|y_0\|_1 + \|y_1\|_1 + \|y_2\|_1 + \|y_3\|_1).$$

Minimizing over $y_4$ gives the convex program (4) in Theorem 4.1. Once we obtain optimal solution $y^\star$ to problem 4, we can take

$$\begin{cases} w_j^\star = \sqrt{y_j^\star}, \alpha_j^\star = \sqrt{y_j^\star}, b_j^\star = -\sqrt{y_j^\star} x_j \text{ for } j = 1, \ldots, n, \\ w_j^\star = \sqrt{y_j^\star}, \alpha_j^\star = \sqrt{y_j^\star}, b_j^\star = -\sqrt{y_j^\star}(x_{j-n} + \epsilon) \text{ for } j = n + 1, \ldots, 2n, \\ w_j^\star = -\sqrt{y_j^\star}, \alpha_j^\star = \sqrt{y_j^\star}, b_j^\star = \sqrt{y_j^\star} x_{j-2n} \text{ for } j = 2n + 1, \ldots, 3n, \\ w_j^\star = -\sqrt{y_j^\star}, \alpha_j^\star = \sqrt{y_j^\star}, b_j^\star = \sqrt{y_j^\star}(x_{j-3n} - \epsilon) \text{ for } j = 3n + 1, \ldots, 4n, \\ b_0^\star = -\frac{1}{n} 1^T([A_1, A_1, A_2, A_2] y^\star), \end{cases}$$

then score matching objective has the same value as optimal value of convex program (4) as $\epsilon \to 0$, which indicates $p^\star = d^\star$ and the above parameter set is optimal. $\qquad \square$

## A.4 PROOF FOR THEOREM 4.1: NEURAL NETWORK TYPE II

*Proof.* Consider data $x \in \mathbb{R}^n$, then the score matching objective is reduced to

$$p^\star = \min_{w,\alpha,b} \frac{1}{2} \left\| \sum_{j=1}^m |xw_j + 1b_j|\alpha_j + 1b_0 \right\|_2^2 + 1^T \left( \sum_{j=1}^m w_j\alpha_j\text{sign}(xw_j + 1b_j) \right) + \frac{1}{2}\beta \sum_{j=1}^m (w_j^2 + \alpha_j^2).$$

According to Lemma 2 in Pilanci & Ergen (2020), after rescaling, the above problem is equivalent to

$$\min_{\substack{w,\alpha,b \\ |w_j|=1}} \frac{1}{2} \left\| \sum_{j=1}^m |xw_j + 1b_j|\alpha_j + 1b_0 \right\|_2^2 + 1^T \left( \sum_{j=1}^m w_j\alpha_j\text{sign}(xw_j + 1b_j) \right) + \beta \sum_{j=1}^m |\alpha_j|,$$

which can be written as

$$\min_{\substack{w,\alpha,b,r_1,r_2 \\ |w_j|=1}} \quad \frac{1}{2}\|r_1\|_2^2 + 1^T r_2 + \beta \sum_{j=1}^m |\alpha_j|$$

$$\text{s.t.} \quad r_1 = \sum_{j=1}^m |xw_j + 1b_j|\alpha_j + 1b_0 \tag{13}$$

$$r_2 = \sum_{j=1}^m w_j\alpha_j\text{sign}(xw_j + 1b_j).$$

The dual problem of (13) writes

$$d^\star = \max_{z_1,z_2} \min_{\substack{w,\alpha,b,r_1,r_2 \\ |w_j|=1}} \frac{1}{2}\|r_1\|_2^2 + 1^T r_2 + \beta \sum_{j=1}^m |\alpha_j| + z_1^T \left( r_1 - \sum_{j=1}^m |xw_j + 1b_j|\alpha_j - 1b_0 \right)$$

$$+ z_2^T \left( r_2 - \sum_{j=1}^m w_j\alpha_j\text{sign}(xw_j + 1b_j) \right),$$

which is a lower bound of optimal value to the original problem, i.e., $p^\star \geq d^\star$. Minimizing over $r_1$ and $r_2$ gives

$$\max_z \min_{\substack{w,\alpha,b \\ |w_j|=1}} -\frac{1}{2}\|z\|_2^2 + \beta \sum_{j=1}^m |\alpha_j| - z^T \left( \sum_{j=1}^m |xw_j + 1b_j|\alpha_j + 1b_0 \right) + 1^T \sum_{j=1}^m w_j\alpha_j\text{sign}(xw_j + 1b_j).$$

Minimizing over $\alpha_j$ gives

$$\max_z \min_{b_0} -\frac{1}{2}\|z\|_2^2 - b_0 z^T 1$$

$$\text{s.t.} \ |z^T|xw_j + 1b_j| - w_j 1^T\text{sign}(xw_j + 1b_j)| \leq \beta, \quad \forall |w_j| = 1, \forall b_j,$$

which is equivalent to

$$\max_z \min_{b_0} -\frac{1}{2}\|z\|_2^2 - b_0 z^T 1$$

$$\text{s.t.} \begin{cases} |z^T|x + 1b_j| - 1^T\text{sign}(x + 1b_j)| \leq \beta \\ |z^T| - x + 1b_j| + 1^T\text{sign}(-x + 1b_j)| \leq \beta \end{cases} \quad \forall b_j.$$

For the constraints to hold, we must have $z^T 1 = 0$ and $b_j$ takes values over $x_j$'s. Furthermore, since sign is discontinuous at input 0, we add another function $\text{sign}^*$ which takes value $-1$ at input 0 to

cater for the constraints. The above is equivalent to

$$
\max_z \quad -\frac{1}{2}\|z\|_2^2
$$
$$
\text{s.t.} \begin{cases} |z^T|x - 1x_i| - 1^T\text{sign}(x - 1x_i)| \le \beta \\ |z^T|x - 1x_i| - 1^T\text{sign}^*(x - 1x_i)| \le \beta \\ |z^T| - x + 1x_i| + 1^T\text{sign}(-x + 1x_i)| \le \beta \\ |z^T| - x + 1x_i| + 1^T\text{sign}^*(-x + 1x_i)| \le \beta \\ z^T 1 = 0 \end{cases} \quad \forall i = 1, \dots, n. \tag{14}
$$

Since the second constraint overlaps with the third, and the fourth constraint overlaps with the first, (14) is equivalent to

$$
\max_z -\frac{1}{2}\|z\|_2^2
$$
$$
\text{s.t.} \begin{cases} |z^T|x - 1x_i| - 1^T\text{sign}(x - 1x_i)| \le \beta \\ |z^T| - x + 1x_i| + 1^T\text{sign}(-x + 1x_i)| \le \beta \\ z^T 1 = 0 \end{cases} \quad \forall i = 1, \dots, n. \tag{15}
$$

According to Lemma A.2, when $\beta > 1$, the constraints in (15) are strictly feasible, thus Slater's condition holds and the dual problem writes

$$
d^\star = \min_{\substack{z_0,z_1,z_2,z_3,z_4 \\ \text{s.t.} z_0,z_1,z_2,z_3 \ge 0}} \max_z -\frac{1}{2}\|z\|_2^2 + \sum_{i=1}^n z_{0i}\left(z^T|x - 1x_i| - 1^T\text{sign}(x - 1x_i) + \beta\right)
$$
$$
+ \sum_{i=1}^n z_{1i}\left(-z^T|x - 1x_i| + 1^T\text{sign}(x - 1x_i) + \beta\right)
$$
$$
+ \sum_{i=1}^n z_{2i}\left(z^T| - x + 1x_i| + 1^T\text{sign}(-x + 1x_i) + \beta\right)
$$
$$
+ \sum_{i=1}^n z_{3i}\left(-z^T| - x + 1x_i| - 1^T\text{sign}(-x + 1x_i) + \beta\right)
$$
$$
+ z_4 z^T 1,
$$

which is equivalent to

$$
\min_{\substack{z_0,z_1,z_2,z_3,z_4 \\ \text{s.t.} z_0,z_1,z_2,z_3 \ge 0}} \max_z -\frac{1}{2}\|z\|_2^2 + e^T z + f,
$$

where $e = \sum_{i=1}^n z_{0i}|x - 1x_i| - \sum_{i=1}^n z_{1i}|x - 1x_i| + \sum_{i=1}^n z_{2i}| - x + 1x_i| - \sum_{i=1}^n z_{3i}| - x + 1x_i| + 1z_4$ and $f = -\sum_{i=1}^n z_{0i}1^T\text{sign}(x - 1x_i) + \sum_{i=1}^n z_{1i}1^T\text{sign}(x - 1x_i) + \sum_{i=1}^n z_{2i}1^T\text{sign}(-x + 1x_i) - \sum_{i=1}^n z_{3i}1^T\text{sign}(-x + 1x_i) + \beta(\|z_0\|_1 + \|z_1\|_1 + \|z_2\|_1 + \|z_3\|_1)$. Maximizing over $z$ gives

$$
\min_{\substack{z_0,z_1,z_2,z_3,z_4 \\ \text{s.t.} z_0,z_1,z_2,z_3 \ge 0}} \frac{1}{2}\|e\|_2^2 + f.
$$

Simplifying to get

$$
\min_{y_1,y_2,z} \frac{1}{2}\|A_1(y_1 + y_2) + 1z\|_2^2 + 1^T C_1 y_1 - 1^T C_2 y_2 + \beta(\|y_1\|_1 + \|y_2\|_1).
$$

Minimizing over $z$ gives the convex program (4) in Theorem 4.1. Once we obtain optimal solution $y^\star$ to problem 4, we can take

$$
\begin{cases} w_j^\star = \sqrt{y_j^\star}, \alpha_j^\star = \sqrt{y_j^\star}, b_j^\star = -\sqrt{y_j^\star}x_j \text{ for } j = 1, \dots, n, \\ w_j^\star = -\sqrt{y_j^\star}, \alpha_j^\star = \sqrt{y_j^\star}, b_j^\star = \sqrt{y_j^\star}x_{j-n} \text{ for } j = n+1, \dots, 2n, \\ b_0^\star = -\frac{1}{n}1^T([A_1, A_1]y^\star), \end{cases}
$$

then score matching objective has the same value as optimal value of convex program (4), which indicates $p^\star = d^\star$ and the above parameter set is optimal. $\qquad\square$

### A.5 PROOF FOR THEOREM 4.1: NEURAL NETWORK TYPE III

*Proof.* Here we reduce score matching objective including ReLU activation to score matching objective including absolute value activation and exploits results in Theorem 4.1 for neural network type IV. Let $\{w^r, b^r, \alpha^r, v^r\}$ denotes parameter set corresponding to ReLU activation, consider another parameter set $\{w^a, b^a, \alpha^a, v^a\}$ satisfying

$$\begin{cases} \alpha^r = 2\alpha^a, \\ w^r = w^a, \\ b^r = b^a, \\ b_0^r = b_0^a - \frac{1}{2}\sum_{j=1}^m b_j^r \alpha_j^r, \\ v^r = v^a - \frac{1}{2}\sum_{j=1}^m w_j^r \alpha_j^r. \end{cases}$$

Then the score matching objective

$$\min_{w^r, \alpha^r, b^r, v^r} \frac{1}{2}\left\|\sum_{j=1}^m (xw_j^r + 1b_j^r)_+ \alpha_j^r + xv^r + 1b_0^r\right\|_2^2 + 1^T\left(\sum_{j=1}^m w_j^r \alpha_j^r \mathbb{1}\{xw_j^r + 1b_j^r \geq 0\}\right) + nv^r + \frac{\beta}{2}\sum_{j=1}^m (w_j^{r\,2} + \alpha_j^{r\,2})$$

is equivalent to

$$\min_{w^a, \alpha^a, b^a, v^a} \frac{1}{2}\left\|\sum_{j=1}^m |xw_j^a + 1b_j^a|\alpha_j^a + xv^a + 1b_0^a\right\|_2^2 + 1^T\left(\sum_{j=1}^m w_j^a \alpha_j^a \mathrm{sign}(xw_j^a + 1b_j^a)\right) + nv^a + \frac{\beta}{2}\sum_{j=1}^m \left(w_j^{a\,2} + 4\alpha_j^{a\,2}\right).$$

According to Lemma 2 in Pilanci & Ergen (2020), after rescaling, the above problem is equivalent to

$$\min_{\substack{w^a, \alpha^a, b^a, v^a \\ |w_j^a| = 1}} \frac{1}{2}\left\|\sum_{j=1}^m |xw_j^a + 1b_j^a|\alpha_j^a + xv^a + 1b_0^a\right\|_2^2 + 1^T\left(\sum_{j=1}^m w_j^a \alpha_j^a \mathrm{sign}(xw_j^a + 1b_j^a)\right) + nv^a + 2\beta\sum_{j=1}^m |\alpha_j^a|. \tag{16}$$

Following similar analysis as in Appendix A.6 with a different rescaling factor, the optimal solution set to (16) is given by

$$\begin{cases} w_j^{a\star} = \sqrt{2y_j^\star}, \alpha_j^{a\star} = \sqrt{y_j^\star/2}, b_j^{a\star} = -\sqrt{2y_j^\star}x_j \text{ for } j = 1, \dots, n, \\ W_j^{a\star} = -\sqrt{2y_j^\star}, \alpha_j^{a\star} = \sqrt{y_j^\star/2}, b_j^{a\star} = \sqrt{2y_j^\star}x_{j-n} \text{ for } j = n+1, \dots, 2n, \\ v^{a\star} = -(\bar{x}^T A_1 y^\star + n)/\|\bar{x}\|_2^2, \\ b_0^{a\star} = -\frac{1}{n}1^T([A_1', A_1']y^\star + xv^{a\star}), \end{cases}$$

where $A_1'$ is $A_1$ defined in Section 4.1.2 and $y^\star$ is optimal solution to convex program (4). Then the optimal parameter set $\{w^r, b^r, \alpha^r, z^r\}$ is given by

$$\begin{cases} w_j^{r\star} = \sqrt{2y_j^\star}, \alpha_j^{r\star} = \sqrt{2y_j^\star}, b_j^{r\star} = -\sqrt{2y_j^\star}x_j \text{ for } j = 1, \dots, n, \\ w_j^{r\star} = -\sqrt{2y_j^\star}, \alpha_j^{r\star} = \sqrt{2y_j^\star}, b_j^{r\star} = \sqrt{2y_j^\star}x_{j-n} \text{ for } j = n+1, \dots, 2n, \\ v^{r\star} = -(\bar{x}^T A_1 y^\star + n)/\|\bar{x}\|_2^2 - \sum_{j=1}^m w_j^{r\star}\alpha_j^{r\star}/2, \\ b_0^{r\star} = -\frac{1}{n}1^T([A_1', A_1']y^\star + x(-(\bar{x}^T A_1 y^\star + n)/\|\bar{x}\|_2^2)) - \sum_{j=1}^m b_j^{r\star}\alpha_j^{r\star}/2. \end{cases}$$

□

### A.6 PROOF FOR THEOREM 4.1: NEURAL NETWORK TYPE IV

*Proof.* Consider data matrix $x \in \mathbb{R}^n$, then the score matching objective is reduced to

$$p^\star = \min_{w, \alpha, b, v} \frac{1}{2}\left\|\sum_{j=1}^m |xw_j + 1b_j|\alpha_j + xv + 1b_0\right\|_2^2 + 1^T\left(\sum_{j=1}^m w_j \alpha_j \mathrm{sign}(xw_j + 1b_j)\right) + nv + \frac{1}{2}\beta\sum_{j=1}^m (w_j^2 + \alpha_j^2).$$

Following similar analysis as in Appendix A.4, we can derive the dual problem as

$$d^\star = \max_{z_1, z_2} \min_{\substack{w, \alpha, b, v, r_1, r_2 \\ |w_j|=1}} \frac{1}{2}\|r_1\|_2^2 + 1^T r_2 + nv + \beta \sum_{j=1}^m |\alpha_j| + z_1^T \left( r_1 - \sum_{j=1}^m |xw_j + 1b_j|\alpha_j - xv - 1b_0 \right)$$
$$+ z_2^T \left( r_2 - \sum_{j=1}^m w_j \alpha_j \text{sign}(xw_j + 1b_j) \right).$$

which gives a lower bound of $p^\star$. Minimizing over $r_1$ and $r_2$ gives

$$\max_{z_1} \min_{\substack{w, \alpha, b, v \\ |w_j|=1}} -\frac{1}{2}\|z_1\|_2^2 + nv + \beta \sum_{j=1}^m |\alpha_j| - z_1^T \left( \sum_{j=1}^m |xw_j + 1b_j|\alpha_j + xv + 1b_0 \right) + 1^T \sum_{j=1}^m w_j \alpha_j \text{sign}(xw_j + 1b_j).$$

Minimizing over $v$ gives

$$\max_{\substack{z_1 \\ z_1^T x = n}} \min_{\substack{w, \alpha, b \\ |w_j|=1}} -\frac{1}{2}\|z_1\|_2^2 + \beta \sum_{j=1}^m |\alpha_j| - z_1^T \left( \sum_{j=1}^m |xw_j + 1b_j|\alpha_j + 1b_0 \right) + 1^T \sum_{j=1}^m w_j \alpha_j \text{sign}(xw_j + 1b_j).$$

Minimizing over $\alpha_j$ gives

$$\max_z \min_{b_0} -\frac{1}{2}\|z\|_2^2 - b_0 z^T 1$$
$$\text{s.t.} \begin{cases} z^T x = n \\ |z^T|xw_j + 1b_j| - w_j 1^T \text{sign}(xw_j + 1b_j)| \leq \beta, \quad \forall |w_j| = 1, \forall b_j. \end{cases}$$

Following same logic as in Appendix A.4, the above problem is equivalent to

$$\max_z \quad -\frac{1}{2}\|z\|_2^2$$
$$\text{s.t.} \begin{cases} |z^T|x - 1x_i| - 1^T \text{sign}(x - 1x_i)| \leq \beta \\ |z^T| - x + 1x_i| + 1^T \text{sign}(-x + 1x_i)| \leq \beta \\ z^T 1 = 0 \\ z^T x = n \end{cases} \quad \forall i = 1, \dots, n. \tag{17}$$

According to Lemma A.3, when $\beta > 2$, the constraints in (17) are strictly feasible, thus Slater's condition holds and the dual problem writes

$$d^\star = \min_{\substack{z_0, z_1, z_2, z_3, z_4, z_5 \\ \text{s.t.} z_0, z_1, z_2, z_3 \geq 0}} \max_z -\frac{1}{2}\|z\|_2^2 + \sum_{i=1}^n z_{0i} \left( z^T|x - 1x_i| - 1^T \text{sign}(x - 1x_i) + \beta \right)$$
$$+ \sum_{i=1}^n z_{1i} \left( -z^T|x - 1x_i| + 1^T \text{sign}(x - 1x_i) + \beta \right)$$
$$+ \sum_{i=1}^n z_{2i} \left( z^T| - x + 1x_i| + 1^T \text{sign}(-x + 1x_i) + \beta \right)$$
$$+ \sum_{i=1}^n z_{3i} \left( -z^T| - x + 1x_i| - 1^T \text{sign}(-x + 1x_i) + \beta \right)$$
$$+ z_4(z^T x - n) + z_5 z^T 1,$$

which is equivalent to

$$\min_{\substack{z_0, z_1, z_2, z_3, z_4, z_5 \\ \text{s.t.} z_0, z_1, z_2, z_3 \geq 0}} \max_z -\frac{1}{2}\|z\|_2^2 + e^T z + f,$$

where $e = \sum_{i=1}^n z_{0i}|x - 1x_i| - \sum_{i=1}^n z_{1i}|x - 1x_i| + \sum_{i=1}^n z_{2i}| - x + 1x_i| - \sum_{i=1}^n z_{3i}| - x + 1x_i| + xz_4 + 1z_5$ and $f = -\sum_{i=1}^n z_{0i}1^T \text{sign}(x - 1x_i) + \sum_{i=1}^n z_{1i}1^T \text{sign}(x - 1x_i) + \sum_{i=1}^n z_{2i}1^T \text{sign}(-x +$

$1 x_i) - \sum_{i=1}^{n} z_{3i} 1^T \text{sign}(-x + 1 x_i) - z_4 n + \beta(\|z_0\|_1 + \|z_1\|_1 + \|z_2\|_1 + \|z_3\|_1)$. Maximizing over $z$ gives

$$\min_{\substack{z_0, z_1, z_2, z_3, z_4, z_5 \\ \text{s.t.} z_0, z_1, z_2, z_3 \geq 0}} \frac{1}{2} \|e\|_2^2 + f.$$

Simplifying to get

$$\min_{y_0, y_1, y_2, y_3} \frac{1}{2} \left\| A_1'(y_0 + y_1) + x y_2 + 1 y_3 \right\|_2^2 + 1^T C_1 y_0 - 1^T C_2 y_1 + n y_2 + \beta(\|y_1\|_1 + \|y_2\|_1), \tag{18}$$

where $A_1', C_1, C_2$ are as $A_1, C_1, C_2$ defined in Section 4.1.2. Minimizing over $y_3$ gives $y_3 = -1^T (A_1'(y_0 + y_1) + x y_2)/n$ and (18) is reduced to

$$\min_{y_0, y_1, y_2} \frac{1}{2} \left\| \bar{A}_1'(y_0 + y_1) + \bar{x} y_2 \right\|_2^2 + 1^T C_1 y_0 - 1^T C_2 y_1 + n y_2 + \beta(\|y_1\|_1 + \|y_2\|_1),$$

where $\bar{A}_1'$ is as $\bar{A}_1$ defined in Section 4.1.2. Minimizing over $y_2$ gives $y_2 = -\left( \bar{x}^T \bar{A}_1'(y_0 + y_1) + n \right)/\|\bar{x}\|_2^2$ and the above problem is equivalent to the convex program (4) in Theorem 4.1. Once we obtain optimal solution $y^\star$ to problem 4, we can take

$$\begin{cases} w_j^\star = \sqrt{y_j^\star}, \alpha_j^\star = \sqrt{y_j^\star}, b_j^\star = -\sqrt{y_j^\star} x_j \text{ for } j = 1, \ldots, n, \\ w_j^\star = -\sqrt{y_j^\star}, \alpha_j^\star = \sqrt{y_j^\star}, b_j^\star = \sqrt{y_j^\star} x_{j-n} \text{ for } j = n+1, \ldots, 2n, \\ v^\star = -(\bar{x}^T A_1 y^\star + n)/\|\bar{x}\|_2^2, \\ b_0^\star = -\frac{1}{n} 1^T([A_1', A_1'] y^\star + x v^\star), \end{cases}$$

then score matching objective has the same value as optimal value of convex program (4), which indicates $p^\star = d^\star$ and the above parameter set is optimal. $\qquad \square$

## A.7 THEOREM 4.2 PROOF

*Proof.* When $X \in \mathbb{R}^{n \times d}$ for some $d > 1$, the score matching objective can be reduced to

$$p^\star = \min_{u_j, v_j} \sum_{i=1}^{n} \left( \frac{1}{2} \left\| \sum_{j=1}^{m} (X_i u_j)_+ v_j^T \right\|_2^2 + \text{tr} \left( \nabla_{X_i} \left[ \sum_{j=1}^{m} (X_i u_j)_+ v_j^T \right] \right) \right), \tag{19}$$

which can be rewritten as

$$\min_{u_j, v_j} \frac{1}{2} \left\| \sum_{j=1}^{m} (X u_j)_+ v_j^T \right\|_F^2 + 1^T \left( \sum_{j=1}^{m} \mathbb{1}\{X u_j \geq 0\} v_j^T u_j \right). \tag{20}$$

Let $D_j' = \text{diag}\left( \mathbb{1}\{X u_j \geq 0\} \right)$, then problem (20) is equivalent to

$$\min_{u_j, v_j} \frac{1}{2} \left\| \sum_{j=1}^{m} D_j' X u_j v_j^T \right\|_F^2 + \sum_{j=1}^{m} \text{tr}(D_j') v_j^T u_j. \tag{21}$$

Thus

$$p^\star = \min_{\substack{W_j = u_j v_j^T \\ (2 D_j' - I) X u_j \geq 0}} \frac{1}{2} \left\| \sum_{j=1}^{m} D_j' X W_j \right\|_F^2 + \sum_{j=1}^{m} \text{tr}(D_j') \text{tr}(W_j) \tag{22}$$

$$\geq \min_{W_j} \frac{1}{2} \left\| \sum_{j=1}^{P} D_j X W_j \right\|_F^2 + \sum_{j=1}^{P} \text{tr}(D_j) \text{tr}(W_j), \tag{23}$$

where $D_1, \ldots, D_P$ enumerates all possible sign patterns of $\text{diag}\left( \mathbb{1}\{X u \geq 0\} \right)$. Let $W_j^\star$ be the optimal solution to the convex program (6). To reconstruct optimal $\{u_j, v_j\}$ to the original training

problem (19), we first factorize $W_j^\star = \sum_{k=1}^{d} \tilde{u}_{jk}\tilde{v}_{jk}^T$. According to Theorem 3.3 in ((Mishkin et al., 2022)), for any $\{j, k\}$, we can write $\tilde{u}_{jk} = \tilde{u}'_{jk} - \tilde{u}''_{jk}$ such that $\tilde{u}'_{jk}, \tilde{u}''_{jk} \in \mathcal{K}_j$ with $\mathcal{K}_j = \{u \in \mathbb{R}^d : (2D_j - I)Xu \succeq 0\}$. Therefore, when $m \geq 2Pd$, we can set $\{u_j, v_j\}$ to enumerate through $\{\tilde{u}'_{jk}, \tilde{v}_{jk}\}$ and $\{\tilde{u}''_{jk}, -\tilde{v}_{jk}\}$ to achieve optimal value of (6). With absolute value activation, replace $D'_j$ to be $\mathrm{diag}\,(\mathrm{sign}(Xu_j))$ and the constraints in 22 become $W_j = u_j v_j^T, D'_j Xu_j \geq 0$, and $D_1, \ldots, D_P$ enumerates all possible sign patterns of $\mathrm{diag}\,(\mathrm{sign}(Xu))$. $\qquad\square$

## A.8 THEOREM 5.1 PROOF

### A.8.1 PROOF FOR NEURAL NETWORK TYPE I

*Proof.* Consider data matrix $x \in \mathbb{R}^{n \times 1}$, then the score matching objective is reduced to

$$p^\star = \min_{w, \alpha, b} \frac{1}{2} \left\| \sum_{j=1}^{m} (xw_j + 1b_j)_+ \alpha_j + 1b_0 - l \right\|_2^2 + \frac{1}{2}\beta \sum_{j=1}^{m} (w_j^2 + \alpha_j^2).$$

According to Lemma 2 in Pilanci & Ergen (2020), after rescaling, the above problem is equivalent to

$$\min_{\substack{w, \alpha, b \\ |w_j|=1}} \frac{1}{2} \left\| \sum_{j=1}^{m} (xw_j + 1b_j)_+ \alpha_j + 1b_0 - l \right\|_2^2 + \beta \sum_{j=1}^{m} |\alpha_j|,$$

which can be rewritten as

$$\min_{\substack{w, \alpha, b, r \\ |w_j|=1}} \frac{1}{2}\|r\|_2^2 + \beta \sum_{j=1}^{m} |\alpha_j| \tag{24}$$
$$\text{s.t. } r = \sum_{j=1}^{m} (xw_j + 1b_j)_+ \alpha_j + 1b_0 - l.$$

The dual of problem (24) writes

$$d^\star = \max_z -\frac{1}{2}\|z\|_2^2 + z^T l$$
$$\text{s.t. } \begin{cases} |z^T(x - 1x_i)_+| \leq \beta \\ |z^T(-x + 1x_i)_+| \leq \beta \qquad \forall i = 1, \ldots, n. \\ z^T 1 = 0 \end{cases}$$

Note the constraint set is strictly feasible since $z = 0$ always satisfies the constraints, Slater's condition holds and we get the dual problem as

$$d^\star = \min_{\substack{z_0, z_1, z_2, z_3, z_4 \\ \text{s.t.} z_0, z_1, z_2, z_3 \geq 0}} \frac{1}{2}\|e\|_2^2 + f,$$

where $e = \sum_{i=1}^{n} z_{0i}(x - 1x_i)_+ - \sum_{i=1}^{n} z_{1i}(x - 1x_i)_+ + \sum_{i=1}^{n} z_{2i}(-x + 1x_i)_+ - \sum_{i=1}^{n} z_{3i}(-x + 1x_i)_+ + 1z_4 + l$ and $f = \beta(\|z_0\|_1 + \|z_1\|_1 + \|z_2\|_1 + \|z_3\|_1)$. Simplify to get

$$\min_y \frac{1}{2}\|Ay + \bar{l}\|_2^2 + \beta\|y\|_1.$$

Once we obtain optimal solution $y^\star$ to problem (10), we can take

$$\begin{cases} w_j^\star = \sqrt{y_j^\star}, \alpha_j^\star = -\sqrt{y_j^\star}, b_j^\star = -\sqrt{y_j^\star}x_j \text{ for } j = 1, \ldots, n, \\ w_j^\star = -\sqrt{y_j^\star}, \alpha_j^\star = -\sqrt{y_j^\star}, b_j^\star = \sqrt{y_j^\star}x_{j-n} \text{ for } j = n+1, \ldots, 2n, \\ b_0^\star = \frac{1}{n}1^T([A_1, A_2]y^\star + l), \end{cases}$$

then denoising score matching objective has the same value as optimal value of convex program (10), which indicates $p^\star = d^\star$ and the above parameter set is optimal. $\qquad\square$

### A.8.2 PROOF FOR NEURAL NETWORK TYPE II

*Proof.* Consider data matrix $x \in \mathbb{R}^{n \times 1}$, then the score matching objective is reduced to

$$p^\star = \min_{w,\alpha,b} \frac{1}{2} \left\| \sum_{j=1}^m |xw_j + 1b_j|\alpha_j + 1b_0 - l \right\|_2^2 + \frac{1}{2}\beta \sum_{j=1}^m (W_j^2 + \alpha_j^2).$$

According to Lemma 2 in Pilanci & Ergen (2020), after rescaling, the above problem is equivalent to

$$\min_{\substack{w,\alpha,b \\ |w_j|=1}} \frac{1}{2} \left\| \sum_{j=1}^m |xw_j + 1b_j|\alpha_j + 1b_0 - l \right\|_2^2 + \beta \sum_{j=1}^m |\alpha_j|, \tag{25}$$

which can be rewritten as

$$\min_{\substack{w,\alpha,b,r \\ |w_j|=1}} \frac{1}{2}\|r\|_2^2 + \beta \sum_{j=1}^m |\alpha_j|$$

$$\text{s.t. } r = \sum_{j=1}^m |xw_j + 1b_j|\alpha_j + 1b_0 - l. \tag{26}$$

The dual of problem (26) writes

$$d^\star = \max_z \min_{\substack{w,\alpha,b,r \\ |w_j|=1}} \frac{1}{2}\|r\|_2^2 + \beta \sum_{j=1}^m |\alpha_j| + z^T \left( r - \sum_{j=1}^m |xW_j + 1b_j|\alpha_j - 1b_0 + l \right),$$

which gives a lower bound of $p^\star$. Minimizing over $r$ and $\alpha$ gives

$$\max_z \min_{b_0} -\frac{1}{2}\|z\|_2^2 - b_0 z^T 1 + z^T l$$

$$\text{s.t. } |z^T|xw_j + 1b_j|| \leq \beta, \quad \forall |w_j| = 1, \forall b_j.$$

Since $w_j = \pm 1$ and $z^T|xw_j + 1b_j|$ is continuous in $b_j$, the above problem is equivalent to

$$\max_z -\frac{1}{2}\|z\|_2^2 + z^T l$$

$$\text{s.t. } \begin{cases} |z^T|x - 1x_i|| \leq \beta \\ |z^T| - x + 1x_i|| \leq \beta \\ z^T 1 = 0 \end{cases} \quad \forall i = 1, \ldots, n.$$

Since the constraints are satisfied by taking $z = 0$, thus Slater's condition holds and the above problem is equivalent to

$$d^\star = \min_{\substack{z_0,z_1,z_2,z_3,z_4 \\ \text{s.t.} z_0,z_1,z_2,z_3 \geq 0}} \max_z -\frac{1}{2}\|z\|_2^2 + z^T l + \sum_{i=1}^n z_{0i}\left(z^T|x - 1x_i| + \beta\right) + \sum_{i=1}^n z_{1i}\left(-z^T|x - 1x_i| + \beta\right)$$

$$+ \sum_{i=1}^n z_{2i}\left(z^T| - x + 1x_i| + \beta\right) + \sum_{i=1}^n z_{3i}\left(-z^T| - x + 1x_i| + \beta\right) + z_4 z^T 1.$$

Maximizing over $z$ gives

$$\min_{\substack{z_0,z_1,z_2,z_3,z_4 \\ \text{s.t.} z_0,z_1,z_2,z_3 \geq 0}} \frac{1}{2}\|e\|_2^2 + f,$$

where $e = \sum_{i=1}^n z_{0i}|x - 1x_i| - \sum_{i=1}^n z_{1i}|x - 1x_i| + \sum_{i=1}^n z_{2i}| - x + 1x_i| - \sum_{i=1}^n z_{3i}| - x + 1x_i| + 1z_4 + l$ and $f = \beta(\|z_0\|_1 + \|z_1\|_1 + \|z_2\|_1 + \|z_3\|_1)$. Simplifying to get

$$\min_{y,z} \frac{1}{2}\|A_3 y + 1z + l\|_2^2 + \beta\|y\|_1.$$

Minimizing over $z$ gives

$$\min_y \frac{1}{2}\|Ay + \bar{l}\|_2^2 + \beta\|y\|_1.$$

Once we obtain optimal solution $y^\star$ to problem (10), we can take

$$\begin{cases} w_j^\star = \sqrt{y_j^\star}, \alpha_j^\star = -\sqrt{y_j^\star}, b_j^\star = -\sqrt{y_j^\star}x_j \text{ for } j = 1,\dots,n, \\ b_0^\star = \frac{1}{n}1^T(A_3y^\star + l), \end{cases}$$

then denoising score matching objective has the same value as optimal value of convex program (10), which indicates $p^\star = d^\star$ and the above parameter set is optimal.

$\square$

### A.8.3 PROOF FOR NEURAL NETWORK TYPE III

*Proof.* Following proof logic in Appendix A.5, let $\{w^r, b^r, \alpha^r, v^r\}$ denotes parameter set corresponding to ReLU activation with skip connection, consider another parameter set $\{w^a, b^a, \alpha^a, z^a\}$ satisfying

$$\begin{cases} \alpha^r = 2\alpha^a, \\ w^r = w^a, \\ b^r = b^a, \\ b_0^r = b_0^a - \frac{1}{2}\sum_{j=1}^m b_j^r\alpha_j^r, \\ v^r = v^a - \frac{1}{2}\sum_{j=1}^m W_j^r\alpha_j^r. \end{cases}$$

Then the denoising score matching objective

$$\min_{w^r,\alpha^r,b^r,v^r} \frac{1}{2}\left\|\sum_{j=1}^m (xw_j^r + 1b_j^r)_+\alpha_j^r + xv^r + 1b_0^r - l\right\|_2^2 + \frac{\beta}{2}\sum_{j=1}^m (w_j^{r\,2} + \alpha_j^{r\,2})$$

is equivalent to

$$\min_{w^a,\alpha^a,b^a,v^a} \frac{1}{2}\left\|\sum_{j=1}^m |xw_j^a + 1b_j^a|\alpha_j^a + xv^a + 1b_0^a - l\right\|_2^2 + \frac{\beta}{2}\sum_{j=1}^m \left(w_j^{a\,2} + 4\alpha_j^{a\,2}\right).$$

According to Lemma 2 in Pilanci & Ergen (2020), after rescaling, the above problem is equivalent to

$$\min_{\substack{w^a,\alpha^a,b^a,v^a \\ |w_j^a|=1}} \frac{1}{2}\left\|\sum_{j=1}^m |xw_j^a + 1b_j^a|\alpha_j^a + xv^a + 1b_0^a - l\right\|_2^2 + 2\beta\sum_{j=1}^m |\alpha_j^a|. \tag{27}$$

Following similar analysis as in Appendix A.8.4 with a different rescaling factor, the optimal solution set to (27) is given by

$$\begin{cases} w_j^{a\star} = \sqrt{2y_j^\star}, \alpha_j^{a\star} = -\sqrt{y_j^\star/2}, b_j^{a\star} = -\sqrt{2y_j^\star}x_j \text{ for } j = 1,\dots,n, \\ w_j^{a\star} = -\sqrt{2y_j^\star}, \alpha_j^{a\star} = -\sqrt{y_j^\star/2}, b_j^{a\star} = \sqrt{2y_j^\star}x_{j-n} \text{ for } j = n+1,\dots,2n, \\ v^{a\star} = \bar{x}^T((I - \frac{1}{n}11^T)A_3y^\star + \bar{l})/\|\bar{x}\|_2^2, \\ b_0^{a\star} = \frac{1}{n}1^T(A_3y^\star - xv^{a\star} + l), \end{cases}$$

where $y^\star$ is optimal solution to convex program (10). Then the optimal parameter set $\{w^r, b^r, \alpha^r, v^r\}$ is given by

$$\begin{cases} w_j^{r\star} = \sqrt{2y_j^\star}, \alpha_j^{r\star} = -\sqrt{2y_j^\star}, b_j^{r\star} = -\sqrt{2y_j^\star}x_j \text{ for } j = 1,\dots,n, \\ w_j^{r\star} = -\sqrt{2y_j^\star}, \alpha_j^{r\star} = -\sqrt{2y_j^\star}, b_j^{r\star} = \sqrt{2y_j^\star}x_{j-n} \text{ for } j = n+1,\dots,2n, \\ v^{r\star} = \bar{x}^T((I - \frac{1}{n}11^T)A_3y^\star + \bar{l})/\|\bar{x}\|_2^2 - \sum_{j=1}^m w_j^{r\star}\alpha_j^{r\star}/2, \\ b_0^{r\star} = \frac{1}{n}1^T(A_3y^\star - x(\bar{x}^T((I - \frac{1}{n}11^T)A_3y^\star + \bar{l})/\|\bar{x}\|_2^2) + l) - \sum_{j=1}^m b_j^{r\star}\alpha_j^{r\star}/2. \end{cases}$$

$\square$

### A.8.4 PROOF FOR NEURAL NETWORK TYPE IV

*Proof.* Consider data matrix $x \in \mathbb{R}^{n \times 1}$, then the score matching objective is reduced to

$$p^\star = \min_{w,\alpha,b,v} \frac{1}{2} \left\| \sum_{j=1}^m |xw_j + 1b_j|\alpha_j + xv + 1b_0 - l \right\|_2^2 + \frac{1}{2}\beta \sum_{j=1}^m (w_j^2 + \alpha_j^2).$$

After applying the rescaling strategy as in Pilanci & Ergen (2020), the above problem is equivalent to

$$\min_{\substack{w,\alpha,b,v,r \\ |w_j|=1}} \frac{1}{2}\|r\|_2^2 + \beta \sum_{j=1}^m |\alpha_j|$$

$$\text{s.t. } r = \sum_{j=1}^m |xw_j + 1b_j|\alpha_j + xv + 1b_0 - l.$$

The dual problem is given by

$$d^\star = \max_z -\frac{1}{2}\|z\|_2^2 + z^T l$$

$$\text{s.t. } \begin{cases} |z^T|x - 1x_i\| \le \beta \\ |z^T| - x + 1x_i\| \le \beta \\ z^T 1 = 0 \\ z^T x = 0 \end{cases} \qquad \forall i = 1, \dots, n,$$

which gives a lower bound of $p^\star$. Note the constraint set is satisfied by taking $z = 0$, thus Slater's condition holds and the dual problem is given by

$$d^\star = \min_{y_1,y_2,y_3} \frac{1}{2}\|A_3 y_1 + x y_2 + 1 y_3 + l\|_2^2 + \beta\|y_1\|_1.$$

Minimizing over $y_2$ and $y_3$ gives the optimal $y_3^\star = -1^T(A_3 y_1 + x y_2 + l)/n$ and $y_2^\star = -\bar{x}^T((I - \frac{1}{n}11^T)A_3 y_1 + \bar{l})/\|\bar{x}\|_2^2$ and the above problem is equivalent to

$$\min_{y_1} \frac{1}{2}\|A y_1 + b\|_2^2 + \beta\|y_1\|_1.$$

Once we obtain optimal solution $y^\star$ to problem (10), we can take

$$\begin{cases} w_j^\star = \sqrt{y_j^\star}, \alpha_j^\star = -\sqrt{y_j^\star}, b_j^\star = -\sqrt{y_j^\star}x_j \text{ for } j = 1, \dots, n, \\ v^\star = \bar{x}^T((I - \frac{1}{n}11^T)A_3 y^\star + \bar{l})/\|\bar{x}\|_2^2, \\ b_0^\star = \frac{1}{n}1^T(A_3 y^\star - x v^\star + l), \end{cases}$$

then denoising score matching objective has the same value as optimal value of convex program (10), which indicates $p^\star = d^\star$ and the above parameter set is optimal.

□

### A.9 THEOREM 5.2 PROOF

*Proof.* When $X \in \mathbb{R}^{n \times d}$ for some $d > 1$, when $\beta = 0$, the score matching objective can be reduced to

$$p^\star = \min_{u_j,v_j} \sum_{i=1}^n \frac{1}{2} \left\| \sum_{j=1}^m (X_i u_j)_+ v_j^T - L_i \right\|_2^2,$$

which can be rewritten as

$$\min_{u_j,v_j} \frac{1}{2} \left\| \sum_{j=1}^m (X u_j)_+ v_j^T - Y \right\|_F^2. \tag{28}$$

Let $D'_j = \text{diag}\,(\mathbb{1}\{Xu_j \geq 0\})$, then problem (28) is equivalent to

$$\min_{u_j, v_j} \frac{1}{2} \left\| \sum_{j=1}^{m} D'_j Xu_j v_j^T - Y \right\|_F^2.$$

Therefore,

$$p^\star = \min_{\substack{W_j = u_j v_j^T \\ (2D'_j - I)Xu_j \geq 0}} \frac{1}{2} \left\| \sum_{j=1}^{m} D'_j XW_j - Y \right\|_F^2$$

$$\geq \min_{W_j} \frac{1}{2} \left\| \sum_{j=1}^{P} D_j XW_j - Y \right\|_F^2,$$

where $D_1, \ldots, D_P$ enumerates all possible sign patterns of $\text{diag}\,(\mathbb{1}\{Xu \geq 0\})$. The construction of optimal parameter set follows Appendix A.7. With absolute value activation, replace $D'_j$ to be $\text{diag}\,(\text{sign}(Xu_j))$ and the constraints in 22 become $W_j = u_j v_j^T, D'_j Xu_j \geq 0$, and $D_1, \ldots, D_P$ enumerates all possible sign patterns of $\text{diag}\,(\text{sign}(Xu))$. $\qquad\square$

### A.10 PROOF OF OPTIMALITY CONDITION IN SECTION 4.1.1

*Proof.* The optimality condition for convex program (4) is

$$0 \in A^T Ay + b + \beta\theta_1, \tag{29}$$

where $\theta_1 \in \partial\|y\|_1$. To show $y^\star$ satisfies optimality condition (29), let $a_i$ denote the $i$th column of $A$. Check the first entry,

$$a_1^T Ay + b_1 + \beta(-1) = nvy_1^\star - nvy_{3n}^\star + n - \beta = 0.$$

Check the $3n$th entry,

$$a_{3n}^T Ay + b_{3n} + \beta = -nvy_1^\star + nvy_{3n}^\star - n + \beta = 0.$$

For $j$th entry with $j \notin \{1, 3n\}$, note

$$|a_j^T Ay + b_j|$$
$$= |a_j^T (a_1 y_1^\star + a_{3n} y_{3n}^\star) + b_j|$$
$$= |a_j^T (a_1 y_1^\star - a_1 y_{3n}^\star) + b_j|$$
$$= \left| \frac{\beta - n}{nv} a_j^T a_1 + b_j \right|.$$

Since $|b_j| \leq n - 1$, by continuity, $|a_j^T Ay + b_j| \leq \beta$ should hold as we decrease $\beta$ a little further to threshold $\beta_2 = \max_{j \notin \{1, 3n\}} |a_j^T Ay + b_j|$. Therefore, $y^\star$ is optimal.

$\qquad\square$

### A.11 PROOF OF OPTIMALITY CONDITION IN SECTION 4.1.2

*Proof.* Assume without loss of generality data points are ordered as $x_1 < \ldots < x_n$, then

$$b = [n, n-2, \cdots, -(n-2), n-2, n-4, \cdots, -n].$$

The optimality condition to the convex program (4) is given by

$$0 \in A^T Ay + b + \beta\theta_1, \tag{30}$$

where $\theta_1 \in \partial\|y\|_1$. To show $y^\star$ satisfies optimality condition (30), let $a_i$ denote the $i$th column of $A$. We check the first entry

$$a_1^T Ay + b_1 + \beta(-1) = nvy_1 - nvy_n + n - \beta = 0.$$

We then check the last entry

$$a_n^T Ay + b_n + \beta = -nvy_1 + nvy_n - n + \beta = 0.$$

For $j$th entry with $1 < j < n$, note

$$
\begin{aligned}
&|a_j^T Ay + b_j| \\
&= |a_j^T(a_1 y_1 + a_n y_n) + b_j| \\
&= |a_j^T(a_1 y_1 - a_1 y_n) + b_j| \\
&= \left| \frac{\beta - n}{nv} a_j^T a_1 + b_j \right|.
\end{aligned}
$$

Since $|b_j| \le n - 2$, by continuity, $|a_j^T Ay + b_j| \le \beta$ should hold as we decrease $\beta$ a little further to some threshold $\beta_1 = \max_{j \notin \{1,n\}} |a_j^T Ay + b_j|$. Therefore, $y^\star$ satisfies (30). $\qquad\square$

### A.12 PROOF FOR CONVERGENCE THEOREMS

#### A.12.1 PROOF FOR THEOREM 6.1

*Proof.* When $s_\theta$ is of neural network type III and IV and $\beta \ge \|b\|_\infty$, the predicted score function is linear with slope $-\frac{1}{v}$ and interception $\mu$ as analyzed in Appendix A.2 and Section 4.1.3, which corresponds to Gaussian distribution with mean $\mu$ and variance $v$. Then since the integrated score function is strongly concave, Theorem 6.1 follows Theorem 4.3.3 in (Chewi, 2023). $\qquad\square$

#### A.12.2 PROOF FOR THEOREM 6.2

*Proof.* When $s_\theta$ is of neural network type II and corresponds to the min-norm solution to the corresponding convex program (4) and $\beta_2 < \beta \le n$, the predicted score function is given by

$$
\begin{cases}
\hat{y} = \frac{\beta - n}{nv}(\hat{x} - \mu), & x_1 \le \hat{x} \le x_n \\
\hat{y} = \frac{\beta - n}{nv}(x_1 - \mu), & \hat{x} < x_1 \\
\hat{y} = \frac{\beta - n}{nv}(x_n - \mu), & \hat{x} > x_n
\end{cases}
$$

as analyzed in Section 4.1.2. Since the score function is bounded by $L = (n - \beta)\max(|x_1 - \mu|, |x_n - \mu|)/nv$, Theorem 6.2 follows Theorem 4.3.9 in (Chewi, 2023). Note Theorem 4.3.3 in (Chewi, 2023) here since the predicted score function is not smooth. $\qquad\square$

### A.13 ANNEALED LANGEVINE SAMPLING

Here we outline the annealed Langevine sampling procedure below,

---

**Algorithm 3** Annealed Langevine Sampling

---

    **Initialize:** $\epsilon_0, \sigma_1, \ldots, \sigma_L, x^0 \sim \mu_0(x)$
    **for** $i = 1, 2, ..., L$ **do**
      $\epsilon_i = \epsilon_0 \sigma_i^2 / \sigma_L^2$
      **for** $t = 1, 2, ..., T$ **do**
        $z^t \sim \mathcal{N}(0, 1)$
        $x^t \leftarrow x^{t-1} + \frac{\epsilon_i}{2} s_{\theta i}(x^{t-1}) + \sqrt{\epsilon_i} z^t$
      **end for**
    **end for**

---

where $s_{\theta i}$ for $i = 1, 2, \ldots, L$ are neural networks trained for denoising score matching with different noise scales.

### A.14 SUPPLEMENTARY SIMULATION RESULTS

In this section, we give more simulation results besides those discussed in main text in Section 7. In Section A.14.1 we show simulation results for score matching tasks with more neural network types and in section A.14.2 we show simulation results for denoising score matching tasks with more neural network types.

### A.14.1    SCORE MATCHING SIMULATION

Figure 5,6, and 7 below show simulation results for score matching tasks with neural network type II, III, IV respectively. In Figure 5, we use the same data samples and training parameters as explained in Section 7. The gap between non-convex training loss and reconstructed optimal neural network loss can be due to the non-smoothness threshold function in the score matching training objective. The score prediction in the middle plot is aligned with our theoretic result in Section 4.1.2 and the sampling histogram in the right plot is consistent with the score prediction. In Figure 6 and 7, we set the weight decay parameter to $\beta = \|b\|_\infty + 1$.[4] Data samples and the rest training parameters are the same as in Section 7. Interestingly, the gap between non-convex training loss and reconstructed optimal neural network loss is minor compared to Figure 5. The predicted score functions are linear with slope close to minus one over sample variance and interception close to sample mean, i.e., $-1$ and $0$ respectively. This is aligned with our theoretic result in Section 4.1.3 and Appendix A.2. The sampling histograms are consistent with the score prediction.

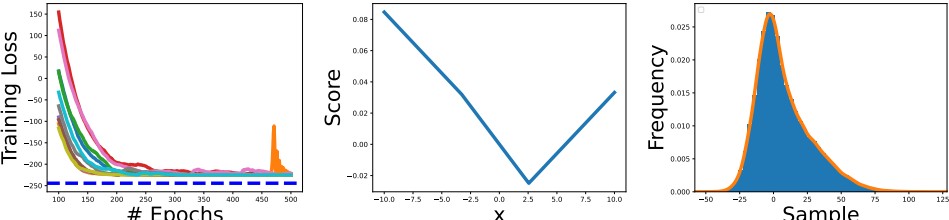

Figure 5: Simulation results for score matching tasks with type II neural network. The left plot shows training loss where the dashed blue line indicates loss of neural network reconstructed from convex program (4). The middle plot shows score prediction from reconstructed optimal neural network. The right plot shows sampling histogram via Langevine process with reconstructed optimal neural network as score estimator. The ground truth distribution is standard Gaussian.

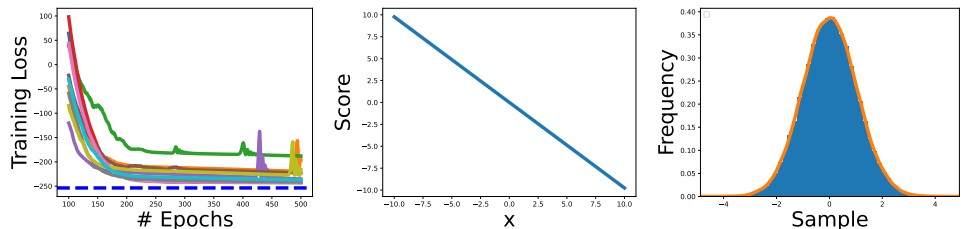

Figure 6: Simulation results for score matching tasks with type III neural network. The left plot shows training loss where the dashed blue line indicates loss of neural network reconstructed from convex program (4). The middle plot shows score prediction from reconstructed optimal neural network. The right plot shows sampling histogram via Langevine process with reconstructed optimal neural network as score estimator. The ground truth distribution is standard Gaussian.

### A.14.2    DENOISING SCORE MATCHING SIMULATION

Figure 8,9, and 10 below show simulation results for denoising score matching tasks for neural network type II, III, IV respectively. We use the same simulation parameters as described in Section 7. The left plots in these three figures show the training loss where the dashed blue line is the objective value obtained by optimal neural network reconstructed from our derived convex program (10). The gap between non-convex training loss and reconstructed optimal neural network loss indicates that our convex program solves the training problem globally. The middle plots in these three figures show sampling histogram via annealed Langevine process with non-convex trained neural network as score estimator. Since the ground truth distribution is standard Gaussian, the

---

[4]see Section 4 for definition of $b$.

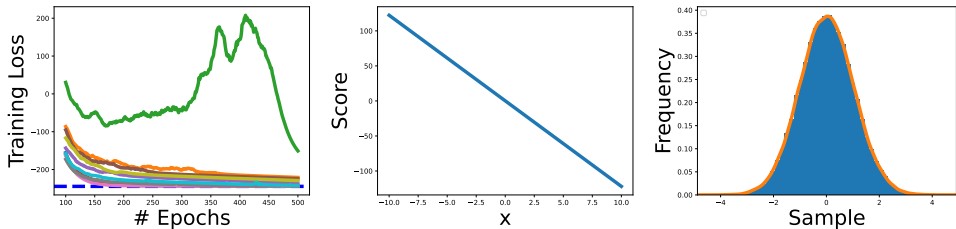

Figure 7: Simulation results for score matching tasks with type IV neural network. The left plot shows training loss where the dashed blue line indicates loss of neural network reconstructed from convex program (4). The middle plot shows score prediction from reconstructed optimal neural network. The right plot shows sampling histogram via Langevine process with reconstructed optimal neural network as score estimator. The ground truth distribution is standard Gaussian.

sample results are not ideal and this is likely caused by that the non-convex trained neural network is not optimal thus the score prediction is not accurate for sampling. The right plot in these three figures show sampling histogram via annealed Langevine process with reconstructed optimal neural network as score estimator. This time the distribution histogram is aligned with standard Gaussian, which indicates the superiority of our convex program used for neural network training.

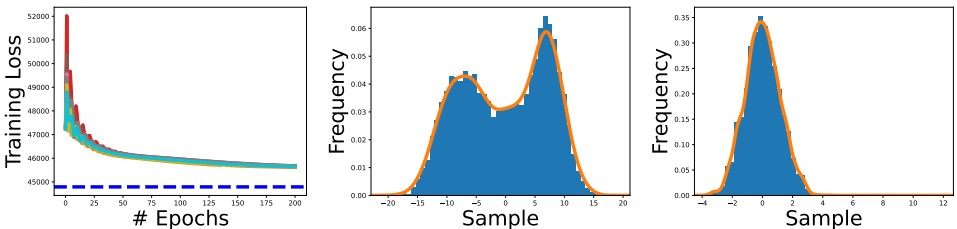

Figure 8: Simulation results for denoising score matching tasks with type II neural network. The left plot shows training loss where the dashed blue line indicates loss of neural network reconstructed from convex program (10). The middle plot shows sampling histogram via annealed Langevine process with non-convex trained neural network as score predictor. The right plot shows sampling histogram via annealed Langevine process with reconstructed optimal neural network as score predictor. The ground truth distribution is standard Gaussian, which is recovered by our model.

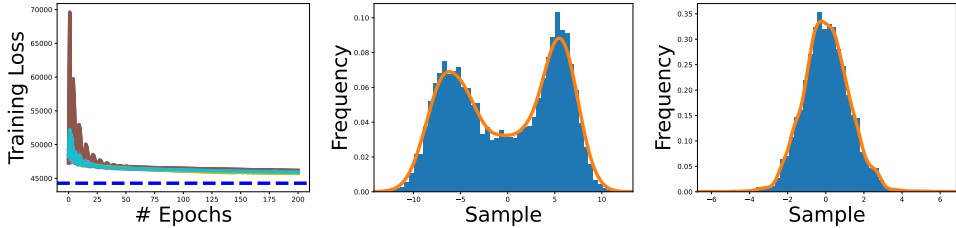

Figure 9: Simulation results for denoising score matching tasks with type III neural network. The left plot shows training loss where the dashed blue line indicates loss of neural network reconstructed from convex program (10). The middle plot shows sampling histogram via annealed Langevine process with non-convex trained neural network as score predictor. The right plot shows sampling histogram via annealed Langevine process with reconstructed optimal neural network as score predictor. The ground truth distribution is standard Gaussian, which is recovered by our model.

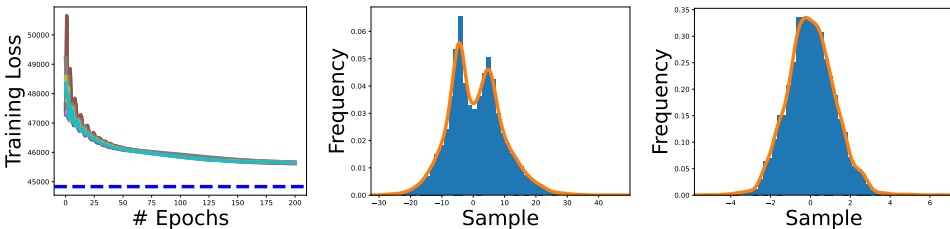

Figure 10: Simulation results for denoising score matching tasks with type IV neural network. The left plot shows training loss where the dashed blue line indicates loss of neural network reconstructed from convex program (10). The middle plot shows sampling histogram via annealed Langevine process with non-convex trained neural network as score predictor. The right plot shows sampling histogram via annealed Langevine process with reconstructed optimal neural network as score predictor. The ground truth distribution is standard Gaussian, which is recovered by our model.

