# OpenReview forum: "Analyzing Neural Network Based Generative Diffusion Models via Convexification"
_ICLR.cc/2024/Conference — Submitted to ICLR 2024_

### Official Review · Reviewer_wVrH · 2023-10-29

**Soundness:** 3 good
**Presentation:** 2 fair
**Contribution:** 3 good
**Rating:** 6
**Confidence:** 4

**Summary:**

The paper studies neural network-based score matching objective. For univariate data, the authors show that the regularized SM can be formulated as a convex optimization problem. The optimal solution to the convex programming can be explicitly obtained and used to recover the neural network parameters. The results are then extended to the multivariate data case. The authors also study the convex optimization problems associated with the regularized denoising score matching loss under both univariate and multivariate data inputs. Finally, the authors investigate the convergence of Langevine MC with the score estimator obtained from the above convex formulation. Numerical experiments are conducted to verify the theoretical results.

**Strengths:**

1. The authors prove that regularized SM and DSM can be viewed as a convex optimization even using neural score estimators. The formulation is obtained under both univariate and multivariate data inputs.
2. The authors study the convergence of Langevin MC with the score estimator obtained from the convex formulation.
3. Numerical experiments are organized to evaluate the performance of score estimator.

**Weaknesses:**

1. The paper considers the regularized score matching loss rather than the standard one.
2. The paper studies the convergence of Langevine dynamics rather than the more widely used backward process in DDPM papers.
3. The Gaussian data distribution in experiments look too simple compared to the ones in practice.

**Questions:**

1. I am curious how the convexity arises. Is it because of the regularization term in the SM and DSM objectives?
2. In many convergence theory, we hope to have a score estimator close to the ground truth score function in $L^2$ sense. Although the optimization objective is derived, it is not clear whether the optimal solution can have good generalization ability in population loss.
3. For Gaussian data distribution, the score function is actually linear. I am surprised in Figure 4 that the non-convex neural estimator trained by DSM cannot recover the ground truth score function. Could you provide some intuition?
4. The writing can be significantly improved. I suggest adding more interpretations after each theoretical result.

---

> ### Author Response · Authors · 2023-11-21
>
> Thanks for the reviewer's comments, here are our replies to the questions:
>
> Weaknesses (1):
>
> Firstly, our convex program for the DSM objective holds for any positive weight decay parameter which can be taken to be very small and zero in the limit. Our convex program for the SM objective requires weight decay parameter to be greater than certain small threshold, which is usually necessary for the SM objective to have finite optimal value. We illustrate this via a small example below.  Consider for example we have only one data point and one hidden neuron, then objective function for NN with ReLU activation and no skip connection would be (see the general form in Appendix A.3)
> \begin{equation*}\begin{aligned}\frac{1}{2}((xw+b)_+\alpha+b_0)^2+w\alpha\mathbb{1}(xw+b\geq 0)+\frac{\beta}{2}(w^2+\alpha^2).\end{aligned}\end{equation*}
> WLOG consider data $x=1,$ then when weight decay $0\leq \beta<1,$ set $b=-w+\sqrt{1-\beta}$ and $b_0=0$ above, we get
> \begin{equation*}
> \begin{aligned}
> \frac{1-\beta}{2}\alpha^2+w\alpha+\frac{\beta}{2}(w^2+\alpha^2)
> \end{aligned}
> \end{equation*}
> Then we can set $\alpha=-w$ and the above expression becomes $(\beta-1)w^2/2$. Thus the objective goes to $-\infty$ when $w$ goes to $\infty$. Note in Theorem 4.1, we requires $\beta>1$, and we consider this reasonable.
>
> Weaknesses (2):
>
> One of our purpose is to show the convexification bypasses some conventional computation difficulties in optimizing SM objective, we consider Langevin dynamics since there is no denoising score matching. We discuss and use the annealed Langevin sampling in our experiments. The annealed Langevin strategy is employed together with denoising score matching for multiple noise levels in Algorithm 3 and then used in our simulation result, see Figure 4 for example.
>
> Weaknesses (3):
>
> The simulation in Figure 3 is used to verify our theoretic results in Section 4.1.1. We explain that no matter what the ground truth data distribution is, when weight decay parameter $\beta$ lies in certain  range, the predicted score function  would always be the same and can be parameterized by sample mean and sample variance. Here we simply pick the ground truth data distribution to be Gaussian. We have added more simulation results with more complex data distributions in the doc and note our theory holds through all these cases. For simulation in Figure 4, the middle plot shows sampling result with non-convex NN solution as score predictor. The sample distribution is nowhere near to standard Gaussian while the sample result with convex score predictor on the right is close to standard Gaussian. We feel in this case, simple Gaussian distribution is enough to reveal the superiority of our convex score predictor compared to its non-convex counterpart.
>
> Questions (1):
>
> This is due to the hidden convexity in ReLU neural networks. Essentially, there exists an embedding of the non-convex problem to a higher-dimensional convex problem. Moreover, the solutions can be mapped between them. However, our results were not known before since the specific form of SM and DSM objectives are quite different from standard supervised learning objectives.
>
> The convexification is not due to the regularization term since the regularization can be taken zero in the limit. However, the weight regularization takes the form of a convex regularizer ($\ell_1$) in the convexified problem.
>
> Questions (2):
>
> In this paper, we mostly focus on the optimization aspects of the SM and DSM objectives and constructing a more transparent convex model equivalent to those. The generalization error can be analyzed based on our results.
>
> Questions (3):
>
> In Figure 4, we do DSM with annealed Langevin sampling, so the neural network is trained to estimate the score function of the noisy data distribution corresponding to multiple noise level. The poor sample result likely indicates that two-layer NN of the form we consider is not good at predicting some of the noisy data distribution and the annealed Langevin sampling with the predicted score function then fails to produce standard Gaussian samples.
>
>
> Questions (4):
>
> Thanks for the suggestions, we will make further improvements regarding to the writing.

---

> > ### Comment · Reviewer_wVrH · 2023-11-22
> > **Response to the rebutttal**
> >
> > Thanks for your efforts. I appreciate your time on the revision. My questions are well-addressed, and I believe my original evaluation is fair.

---

### Official Review · Reviewer_nxW6 · 2023-10-31

**Soundness:** 3 good
**Presentation:** 2 fair
**Contribution:** 2 fair
**Rating:** 5
**Confidence:** 4

**Summary:**

This paper analyzed the score-matching and denoting score-matching objective with two-layer neural networks as the fitting function. For univariate training data, they show that such optimization problems can be reformulated as convex optimization. They establish the Langevin dynamics converges to Gaussian or Gaussian-Laplace distribution. The result is extended to high-dimensional data.

**Strengths:**

This is the first paper that proposes a convex formulation for the score matching with two-layer neural networks. It provides concrete theoretical guarantees.

**Weaknesses:**

1. It seems unnatural to add a regularization term along with the SM or DSM objective function. By adding the regularization term, the minimizer of the SM/DSM objective function (population version) will not be the intended score function, so the generated samples from such diffusion models will not follow the distribution of the training data. Indeed, Theorem 6.1 and 6.2 show that the obtained sample will follow Gaussian/Gaussian-Laplace distribution, no matter which distribution the training data is from. This defeats the original purpose of diffusion generative modeling.
2. The convex program in 1 dimension is efficiently solvable (the regularized objective). However, in high dimensions, the convex objective function involves combinatorial sums, which is not efficiently computable.

Minor:
In the statement of Theorem 4.2, $D_j = D_j$ when ... and $D_j = 2 D_j - 1$ when ... is not a serious statement.

**Questions:**

Could you respond to the two weaknesses above?

---

> ### Author Response · Authors · 2023-11-21
>
> Thanks for the reviewer's comments, here are our replies to the questions:
>
> Weaknesses (1):
>
> Firstly, our convex program for the DSM objective holds for any positive weight decay parameter which can be taken to be very small and zero in the limit. Our convex program for the SM objective requires weight decay parameter to be greater than certain small threshold, which is usually necessary for the SM objective to have finite optimal value. We illustrate this via a small example. Consider for example we have only one data point and one hidden neuron, then objective function for NN with relu activation and no skip coNNection would be (see the general form in Appendix A.3)
> \begin{equation*}\begin{aligned}\frac{1}{2}((xw+b)_+\alpha+b_0)^2+w\alpha\mathbb{1}(xw+b\geq 0)+\frac{\beta}{2}(w^2+\alpha^2).\end{aligned}\end{equation*}
> WLOG consider data $x=1,$ then when weight decay $0\leq \beta<1,$ set $b=-w+\sqrt{1-\beta}$ and $b_0=0$ above, we get
> \begin{equation*}
> \begin{aligned}
> \frac{1-\beta}{2}\alpha^2+w\alpha+\frac{\beta}{2}(w^2+\alpha^2)
> \end{aligned}
> \end{equation*}
> Then we can set $\alpha=-w$ and the above expression becomes $(\beta-1)w^2/2$. Thus the objective goes to $-\infty$ when $w$ goes to $\infty$. Note in Theorem 4.1, we requires $\beta>1$, and we consider this reasonable.
> For DSM objective, we require regularization $\beta>0$ for the strong duality to hold and $\beta$ can be set very small to mimic the zero weight decay case.
>
> Theorem 6.1 and 6.2 discuss some large weight decay regime where we are able to solve the convex program analytically and discover Gaussian/Laplacian mixture. For general weight decay,  once an optimal solution $y^\star$ to the corresponding convex program is derived, the predicted score function is just $\sum y^\star_i\sigma(x-x_i)+b^\star$ where $b^\star$ can be computed from $y^\star$. We update our doc with the following general sampling result. For ReLU activaton, similar result holds by replacing sign function with indicator function.
>
> For NN with absolute value activation, note once the optimal $y^\star$ to the convex program is known, the optimal score function is simply
> \begin{equation*}
> \begin{aligned}
> s(x)=\sum_{i=1}^n (y_{i}^\star+y_{n+i}^\star) |x-x_i|+b^\star
> \end{aligned}
> \end{equation*}
> Integrating the above score function, we obtain
> \begin{equation*}
> \begin{aligned}
> \int s(x)dx=\frac{1}{2}\sum_{i=1}^n (y_{i}^\star+y_{n+i}^\star)(x-x_i)^2\text{sign}(x-x_i)
> +b^\star x
> \end{aligned}
> \end{equation*}
> The corresponding probability density is proportional to
> \begin{equation*}
> \begin{aligned}
> \exp(\int s(x)dx)=\exp(\frac{1}{2}\sum_{i=1}^n (y_{i}^\star+y_{n+i}^\star)(x-x_i)^2\text{sign}(x-x_i)+b^\star x).
> \end{aligned}
> \end{equation*}
>
> Note that the log-density is piecewise quadratic with breakpoints only at a subset of training points.
>
> Weaknesses (2):
>
> The convex objective function says that we enumerating all sign patterns, denote the number of sign patterns as $P$, and solve a convex program with $P$ variables of dimension $d$ by $d$. Here $D_i$ and $X$ is fixed and we are only optimizing over $W_i$, which contains no combinatorial sums. Note $P\in O(r(n/r)^r)$ where $r$ is the rank of data matrix. In order to guarantee global optimality, the program  takes $O((r(n/r)^{r}d^2)^3)$.  In practice, when $P$ grows large, one can simply sample a subset of sign patterns and solve the convex program with these sub-sampled sign patterns. See Theorem 2.1, Theorem 3.3 and Algorithm 1 in [1].
>
> [1] Aaron Mishkin, Arda Sahiner, and Mert Pilanci. Fast convex optimization for two-layer
> relu networks: Equivalent model classes and cone decompositions, 2022.
>
> Minor:
>
> Thanks for pointing out the ambiguity, here we are saying the coefficient matrix $D_j$ in Theorem 4.2 is equal to sign pattern $D_i$ explained at the beginning of Section 4.2 and $D_j=2D_i-I$ for absolute value activation. We've updated the doc to reflect this.

---

### Official Review · Reviewer_Nb2h · 2023-11-01

**Soundness:** 2 fair
**Presentation:** 2 fair
**Contribution:** 1 poor
**Rating:** 3
**Confidence:** 4

**Summary:**

This paper studies the score-matching and denoising score-matching objective functions used to learn the scores of distributions for diffusion models. The focus is on understanding what the vanilla score-matching objective with weight decay does for a two-layer neural network. The authors show that for a certain weight decay parameter regime, the problem can be cast as a convex optimization problem, and the global minimum always corresponds to a distribution close in KL to either a Gaussian or a Gaussian-Laplace distribution. The authors also show that under this regime, the Langevin algorithm can sample from the learned distribution.

**Strengths:**

- This paper attempts to study why score matching for two-layer neural networks can learn the score, which is an important and relevant question.

**Weaknesses:**

- This paper is not well-motivated. The vanilla score-matching objective is not used in practice, and the convex program is not used to train the neural net in practice. So it's not clear why this particular problem is being studied.
- This paper studies a particular weight decay regime, in which the distribution converges to either a Gaussian or Gaussian-Laplace. However, in practice, there is no weight decay. Furthermore, the interesting thing about diffusion models is that they *can* learn more complicated distributions. The simplest example is the mixture of Gaussians, which *even* two-layer neural networks can represent the score of, and which this weight decay regime fails to capture. In practice, of course, the distributions being learned are far more interesting than a Gaussian-Laplace, or even a mixture of Gaussians, and this paper fails to explain any of this.
- There is a small section studying the denoising score-matching objective (the one actually used in practice), but it seems to again be in the uninteresting weight decay regime (this section is very short, and the results are difficult to interpret).
- More generally, the presentation is difficult to follow, and things are not well motivated. For instance thereom 4.1 is stated, but no intuition is provided for what it means. The matrices/vectors in the theorem are described in subsequent sections, but again, not much motivation/intuition is provided. It seems like a lot of this should just be pushed to the appendix.
- The Langevin algorithm is studied for sampling, while in practice, annealed Langevin is used after learning the score at multiple noise levels. None of this is explored in this paper. Furthermore, the sampling result itself seems to be a consequence of classical work, since the distribution learned is always very simple in the weight decay regime studied.

**Questions:**

- What is the motivation for studying this problem?
- Is there anything you can say when there is no weight decay?
- Can you provide more detail in the section studying the denoising score matching objective?

---

> ### Author Response · Authors · 2023-11-21
>
> Thanks for the reviewer's comments, here are our replies to the questions:
>
> Weaknesses (1):
>
> We note that our work resolves the difficulties in direct score matching. Vanilla score-matching objective is not used in practice mainly because of the computational difficulties associated with the  trace of the Jacobian in the objective function. This is due to the second-order nature of its gradients and the pathological discontinuities, e.g., second derivative for common activation functions like ReLU are zero almost everywhere. To tackle this difficulty, methods such as denoising score matching and sliced score matching have been proposed. But there are some drawbacks in these methods, for examples, denoising score matching learns the noisy version of the data distribution which is close to the true data distribution only when noise is small and an iterated process is required; sliced score matching employs a trace estimation trick to achieve asymptotically unbiased estimation of the trace of Jacobian but might be inaccurate in each evaluation. In our work, we showed the convex program of vanilla score-matching objective can be derived and thus training the vanilla objective function is equivalent to solving a convex program, which completely removes the difficulties and makes introducing other additional tricks to approximate unnecessary. More importantly, we derive additional convex programs for denoising score matching, and the simulation shows denoising score matching perform poorly compared to exact score matching (see Figure 4). This might motivate using convex program for training diffusion model in some applications.
>
> Our project is indeed well-motivated. Existing theory works mainly establish convergence of diffusion process when the learned score approximates score of unknown data distribution well, but in reality only empirical approximation is viable due to finite training samples. Current literature falls short in understanding the role of NN approximation error for score-based generative models and it's difficult
> to characterize the distribution from which these models sample in practice. For example, ([1],[2],[3]) show NN-based score-based generative models given finite training data generalizes due to approximation errors introduced by limited NN model capacity and also optimization errors. Our work contributes in understanding what score function NN learns in finite training data regime (see reply to your comment 2 below). Simply speaking, we characterize what two-layer NN exactly learns in finite data regime, there is no optimization error since we are able to solve the convex program exactly.
>
> [1] Jakiw Pidstrigach. Score-based generative models detect manifolds. Advances in Neural Information
> Processing Systems, 35:35852–35865, 2022.
>
> [2] TaeHo Yoon, Joo Young Choi, Sehyun Kwon, and Ernest K. Ryu. Diffusion probabilistic models
> generalize when they fail to memorize. In ICML 2023 Workshop on Structured Probabilistic
> Inference & Generative Modeling, 2023.
>
> [3] Mingyang Yi, Jiacheng Sun, and Zhenguo Li. On the generalization of diffusion model.

---

> > ### Author Response · Authors · 2023-11-21
> >
> > Weaknesses (2):
> >
> > Firstly, with respect to the weight decay, our convex reformulations for DSM hold for any non-zero weight decay (which can be taken to zero in the limit). Our convex program for SM requires weight decay greater than some threshold which is usually necessary for the optimal SM objective value to stay finite, please see explanation below.   In practice, initializing the weights randomly with small magnitude (i.e., Gaussian random initialization with small variance) and running a local optimizer has the same effect penalizing the weight decay term. This was formally proved in [4].
> >
> > In particular, for the vanilla SM objective, we can illustrate via a simple example that regularization is necessary for the optimal objective to be finite. Consider for example we have only one data point and one hidden neuron, then objective function for the NN with ReLU activation and no skip connection would be (see the general form in Appendix A.3)
> > \begin{equation*}
> > \begin{aligned}\frac{1}{2}((xw+b)_+\alpha+b_0)^2+w\alpha\mathbb{1}(xw+b\geq 0)+\frac{\beta}{2}(w^2+\alpha^2).\end{aligned}
> > \end{equation*}
> > WLOG consider data $x=1,$ then when weight decay $0\leq \beta<1,$ set $b=-w+\sqrt{1-\beta}$ and $b_0=0$ above, we get
> > \begin{equation*}
> > \begin{aligned}
> > \frac{1-\beta}{2}\alpha^2+w\alpha+\frac{\beta}{2}(w^2+\alpha^2)
> > \end{aligned}
> > \end{equation*}
> > Then we can set $\alpha=-w$ and the above expression becomes $(\beta-1)w^2/2$. Thus the objective goes to $-\infty$ when $w$ goes to $\infty$. Note in Theorem 4.1, we require $\beta>1$, which is reasonable.
> >
> > Secondly, with respect to the generality, we have some surprising conclusions in certain regimes of weight decay where we are able to solve the convex program analytically and we observe Gaussian/Laplace mixtures. In general, as long as one is able to solve the convex program in Theorem 4.1 and Theorem 5.1 corresponding to the SM objective (with weight decay parameter greater than certain small threshold)  and the DSM objective (with any positive weight decay parameter), once the optimal value $y^\star$  is derived, the predicted score function is just $\sum y^\star_i\sigma(x-x_i)+b^*$ where $b^\star$ can be computed from $y^\star$, which is a piecewise linear function with breakpoints at a subset of data points. We update our doc with the following general sampling result. For ReLU activation, similar result holds by replacing sign function with indicator function.
> >
> > For NN with absolute value activation, note once the optimal $y^\star$ to the convex program is known, the optimal score function is simply
> > \begin{equation*}
> > \begin{aligned}
> > s(x)=\sum_{i=1}^n (y_{i}^\star+y_{n+i}^\star) |x-x_i|+b^\star
> > \end{aligned}
> > \end{equation*}
> > Integrating the above score function, we obtain
> > \begin{equation*}
> > \begin{aligned}\int s(x)dx=\frac{1}{2}\sum_{i=1}^n (y_{i}^\star+y_{n+i}^\star)(x-x_i)^2\text{sign}(x-x_i)
> > +b^\star x\end{aligned}
> > \end{equation*}
> > The corresponding probability density is proportional to
> > \begin{equation*}
> > \begin{aligned}\exp(\int s(x)dx)=\exp(\frac{1}{2}\sum_{i=1}^n (y_{i}^\star+y_{n+i}^\star)(x-x_i)^2\text{sign}(x-x_i)+b^\star x).\end{aligned}
> > \end{equation*}
> > Note that the log-density is piecewise quadratic with breakpoints only at a subset of training points.
> >
> > Thirdly, the claim that two-layer neural networks can represent the score function of a mixture of Gaussians is not exactly true. It's straightforward to show that the score function is not piecewise linear, therefore, ReLU and piecewise linear activations can not exactly model it. However, in the case of well-seperated centers, the score function can be approximated with a piecewise linear function. Alternatively, the approximation error will decay to zero when we increase the number of neurons due to the universal approximation property of two-layer neural networks with suitable activation functions (e.g., ReLU, sigmoid and variants).
> >
> > [4] Y. Wang, M. Pilanci, The convex geometry of backpropagation: Neural network gradient flows converge to extreme points of the dual convex program, ICLR 2022

---

> ### Author Response · Authors · 2023-11-21
>
> Weaknesses (3):
>
> For denoising score-matching, we derive its convex program for weight decay regime $\beta>0$, i.e., as long as the weight decay is non-zero (see Thereom 5.1).  In practice, $\beta$ can be set very small or taken to zero in the limit.
>
> Weaknesses (4):
>
> Thanks for this comment. We will improve the readability in the revision. Since we study the convex program of different neural network types including ReLU and absolute value activation, and with or without skip connections, the convex programs are of similar quadratic form with slightly different coefficient matrices/vectors, so we express the general form in Theorem 4.1 and  specify coefficient matrices/vectors for different neural network types in the following paragraph. We feel that the specific coefficient matrices/vectors are useful for readers to see how the derived convex program is related to the input data so we put them in the main text. Again, our work aims at both  bypassing difficulties in training NN with vanilla SM objective and understanding NN approximation error in the finite training data regime, which is crucial for NN-based score-based generative models (please see our reply to your comment 1).
>
> Weaknesses (5):
>
> In fact, we discuss and use the annealed Langevin sampling in our experiments. The annealed Langevin strategy is employed together with denoising score matching for multiple noise levels in Algorithm 3 and then used in our simulation result, see Figure 4 for example. Our sampling results are enabled by the convex reparameterizations followed by classic convergence result of log concave sampling (see https://chewisinho.github.io/main.pdf). However, they are specialized to the explicit score function obtained by solving the convex programs. Ultimately, our convergence results apply for NN-based score function learned with finite data and we are unaware of any prior work on guarantees for any NN-based learned score function.
>
> Questions (1):
>
> Please see our reply to your comments 1/4/5.
>
> Questions (2):
>
> Please see our reply to your comment 2. To summarize, we show that weight decay is crucial for vanilla SM objective for optimal value to be finite. For DSM objective, our convex program holds for any positive weight decay and can be taken to be zero in the limit. We list some unexpected Gaussian/Gaussian-Laplace results in specific weight decay regimes since we can solve the convex program analytically. But our convex program is general and once the optimal value $y^\star$ and $b^\star$ is derived, the predicted score function is just $\sum y^\star_i\sigma(x-x_i)+b^*$, which is a piecewise linear function with breakpoints at a subset of data points.  Note that the $\ell_1$ regularization encourages the convex program to have a small subset of data points as the breakpoins of the score function.
>
> Questions (3):
>
> Our main result for DSM is the convex program derived in Theorem 5.1. We spent a large portion of the paper on the SM objective since one of our purpose is to show the convexification bypasses some conventional computation difficulties in optimizing the SM objective. For DSM convex program, note it holds for any positive weight decay and it's a simple Lasso problem. Our theoretical and simulation results make the comparison between SM and DSM fully transparent.

---

> > ### Comment · Reviewer_Nb2h · 2023-11-23
> >
> > I am very confused by this paper. At a basic level, the experiments in figure 4 currently don't make sense to me, since of course the denoising score matching with annealed Langevin can be used to learn to sample from a standard gaussian. The description is not at all clear to me currently.
> >
> > I think this paper needs significant rewriting to be understandable. For this reason I maintain my score.

---

> > > ### Author Response · Authors · 2023-11-23
> > >
> > > Thanks for all your comments. Figure 4 shows non-convex trained score function of the specific two-layer form described at the beginning of Section 4. The reason why the sample result is not good might include optimization errors, number of sample step is small, etc. With convex score predictor, the sample is indeed ideal, which demonstrates the superiority of our convex score predictor. We'll add more sampling simulations with more complex data distributions to further show the difference.
> > > The simulation code is attached in case anyone want to try it out.

---

### Official Review · Reviewer_1kxL · 2023-11-09

**Soundness:** 2 fair
**Presentation:** 2 fair
**Contribution:** 2 fair
**Rating:** 5
**Confidence:** 2

**Summary:**

This paper propose to analyse diffusion models by reframing score matching as a convex optimization problem.

**Strengths:**

The score matching is generally viewed as a nonconvex problem, hence the guarantees in this sense for diffusion models are hard to obtain. The paper tries to develop a convex optimization approach, which could significantly improve theoretical understanding of these training procedures.

**Weaknesses:**

The general promise of the paper seems too ambitious, as of course, there is no way (yet) to obtain general convex optimisation formulation. The paper focuses on specific cases and somehow it has to be clarified for full transparency that the results apply to specific cases.

**Questions:**

1- The results of convexification relies on specific structure of the networks that makes sense. However, it is quite obvious that, also, these networks are not what's used in practice. It would be appropriate to update the text to reflect the limitations.

2- The authors also didn't discuss the general difficulty of extending these results to more realistic networks. A discussion on this would be helpful. In fact, ReLU can be replaced with other basic units, would these results hold or easily extended to those cases?

3- While authors mentioned the computational difficulty of score matching, it would be appropriate to discuss the numerical cost of this procedure and have a comparison with score matching in terms of runtimes for similar problems. Is it much more efficient?

Typos

- The word "Langevine" used in many parts of the paper instead of Langevin
- Section 3 can be incorporated as a subsection of Section 1.

---

> ### Author Response · Authors · 2023-11-21
>
> Thanks for the reviewer's comments, here are our replies to the questions:
>
> Weaknesses:
>
> We state clearly that we present a theoretical framework to analyze two-layer NN-based diffusion models in the abstract, this is realized via Theorem 3.1 for univariate data and Theorem 3.2 for multivariate data for score matching objective, and via Theorem 4.1/4.2 for denoising score matching objective. The neural network model we study is presented at the beginning of Section 4, which includes both ReLU and absolute value activation and with or without skip connection. We consider this general enough for two-layer neural network architecture.  In our analysis for predicted score function, we focus on specific weight decay regime where we can solve the convex program analytically and we discover Gaussian/Laplacian mixture. For general weight decay,  once an optimal solution $y^\star$ to the corresponding convex program is derived, the predicted score function is just $\sum y^\star_i\sigma(x-x_i)+b^\star$ where $b^\star$ can be computed from $y^\star$. We update our doc with the following general sampling result. For ReLU activaton, similar result holds by replacing sign function with indicator function.
>
> For NN with absolute value activation, note once the optimal $y^\star$ to the convex program is known, the optimal score function is simply
> \begin{equation*}\begin{aligned}s(x)=\sum_{i=1}^n (y_{i}^\star+y_{n+i}^\star) |x-x_i|+b^\star\end{aligned}\end{equation*}
> Integrating the above score function, we obtain
> \begin{equation*}\begin{aligned}\int s(x)dx=\frac{1}{2}\sum_{i=1}^n (y_{i}^\star+y_{n+i}^\star)(x-x_i)^2\text{sign}(x-x_i)
> +b^\star x\end{aligned}\end{equation*}
> The corresponding probability density is proportional to
> \begin{equation*}\begin{aligned}\exp(\int s(x)dx)=\exp(\frac{1}{2}\sum_{i=1}^n (y_{i}^\star+y_{n+i}^\star)(x-x_i)^2\text{sign}(x-x_i)+b^\star x).\end{aligned}\end{equation*}
>
> Note that the log-density is piecewise quadratic with breakpoints only at a subset of training points.
>
> Questions (1):
>
> Please see our reply for  your comment 1. Our work aims at both revealing the ability of  bypassing conventional difficulties in training NN with vanilla SM objective via convexification and understanding NN approximation error in finite training data regime, which is crucial for NN-based score-based generative models.  Existing theory works mainly establish convergence of diffusion process when the learned score approximates score of unknown data distribution well, but in reality only empirical approximation is viable due to finite training samples. Current literature falls short in understanding the role of NN approximation error for score-based generative models and it's difficult
> to characterize the distribution from which these models sample in practice. For example, ([1],[2],[3]) show NN-based score-based generative models given finite training data generalizes due to approximation errors introduced by limited NN model capacity and also optimization errors. We establish convergence results apply for NN-based score function learned with finite data and we are unaware of any prior work on guarantees for any NN-based learned score function.
>
> [1] Jakiw Pidstrigach. Score-based generative models detect manifolds. Advances in Neural Information
> Processing Systems, 35:35852–35865, 2022.
>
> [2] TaeHo Yoon, Joo Young Choi, Sehyun Kwon, and Ernest K. Ryu. Diffusion probabilistic models
> generalize when they fail to memorize. In ICML 2023 Workshop on Structured Probabilistic
> Inference & Generative Modeling, 2023.
>
> [3] Mingyang Yi, Jiacheng Sun, and Zhenguo Li. On the generalization of diffusion model.
>
> Questions (2):
>
> The current convex reformulation can be extended to any piecewise linear activation, it is not obvious how to extend it to smooth activations such as SiLU/GeLU trivially. However, since our reformulation resolves the differentiability issue of the ReLU activation, and hence resorting to smooth activations might be unnecessary.
>
> It's possible to extend the convex formulations to deeper networks based on recursive hyperplane arrangements (see [4] and [5]). However, our work uncovers many complexities even in two-layer ReLU score models that were not known before.
>
> [4] T. Ergen, M. Pilanci, Path Regularization: A Convexity and Sparsity Inducing Regularization for Parallel ReLU Networks. NeurIPS 2023.
>
> [5] T. Ergen, M. Pilanci, Global Optimality Beyond Two Layers: Training Deep ReLU Networks via Convex Programs.
> ICML 2021.

---

> ### Author Response · Authors · 2023-11-21
>
> Questions (3):
>
> Our convex program in Theorem 3.1 is a standard cone program with multiple of $n$ variables and no constraints  when we have $n$ training points. Therefore, the complexity of solving it with standard interior point methods is $O(n^3)$. Our convex program in Theorem 3.2 is a standard cone program with $Pd^2$ variables where $d$ is the dimension of the training samples and $P$ is the number of samples for the hyperplane arrangement. The complexity is $O((Pd^2)^3)$. In order to guarantee global optimality, the program in Theorem 3.1 takes $O(n^3)$ time and the program in Theorem 3.2 takes $O((r(n/r)^{r}d^2)^3)$ where $r$ is the rank of the training data matrix.
>
> In contrast, applying a gradient method such as (stochastic) gradient descent has no guarantees of convergence or local optimality. At best, one could aim to prove convergence to a stationary point, however, to the best of our knowledge, this is not known for the SM or DSM objectives.
>
> Typos:
>
> Thanks for pointing out the typos, we have made update according to the suggestions.

---

### Meta-Review · Area_Chair_EEPD · 2023-12-04

**Metareview:**

The authors show that in a certain regime, two-layer neural network-based diffusion models can be reframed as a convex optimization problem, which can be efficiently solved. Optimization for score-based diffusion models currently lacks good theoretical guarantees, so this is an important research direction.

However, the reviewers pointed out many restrictions of this work: the theoretical guarantees only give convergence to a Gaussian or a Gaussian-Laplace distribution which is too simple compared to the complex distributions that diffusion models are used to learn (ex., a mixture of gaussians would at least capture some of the complexity), the paper mainly considers the 1-dimensional case and the efficiency in the multi-dimensional case is unclear, and the objective/architecture is different from what is used in practice (which may be acceptable if not for the other shortcomings).

**Justification For Why Not Higher Score:**

The results of the paper are too limited and far from the practical/interesting settings for diffusion models.

**Justification For Why Not Lower Score:**

N/A

---

### Decision · Program_Chairs · 2024-01-16

Reject